# Strontium Sulfite: A New pH-Responsive Inorganic Nanocarrier to Deliver Therapeutic siRNAs to Cancer Cells

**DOI:** 10.3390/pharmaceutics11020089

**Published:** 2019-02-20

**Authors:** Md. Emranul Karim, Jayalaxmi Shetty, Rowshan Ara Islam, Ahsanul Kaiser, Athirah Bakhtiar, Ezharul Hoque Chowdhury

**Affiliations:** 1Jeffrey Cheah School of Medicine and Health Sciences, Monash University Malaysia, Jalan Lagoon Selatan, Bandar Sunway, 47500 Petaling Jaya, Malaysia; karim604306@gmail.com (M.E.K.); jayashetty21@gmail.com (J.S.); Rowshan.Islam@monash.edu (R.A.I.); ahsanul.kaiser@monash.edu (A.K.); 2Faculty of Pharmacy, Mahsa University, 2, Jalan SP 4/4, Bandar Saujana Putra, 42610 Jenjarom, Malaysia; athirahbakhtiar@gmail.com

**Keywords:** inorganic nanoparticles, tumor, strontium sulfite, pH-responsive drug delivery, siRNA, gene therapy, breast cancer

## Abstract

Inorganic nanoparticles hold great potential in the area of precision medicine, particularly for treating cancer owing to their unique physicochemical properties, biocompatibility and improved pharmacokinetics properties compared to their organic counterparts. Here we introduce strontium sulfite nanoparticles as new pH-responsive inorganic nanocarriers for efficient transport of siRNAs into breast cancer cells. We employed the simplest nanoprecipitation method to generate the strontium sulfite nanoparticles (SSNs) and demonstrated the dramatic roles of NaCl and d-glucose in particle growth stabilization in order to produce even smaller nanosize particles (Na-Glc-SSN) with high affinity towards negatively charged siRNA, enabling it to efficiently enter the cancer cells. Moreover, the nanoparticles were found to be degraded with a small drop in pH, suggesting their potential capability to undergo rapid dissolution at endosomal pH so as to release the payload. While these particles were found to be nontoxic to the cells, they showed higher potency in facilitating cancer cell death through intracellular delivery and release of oncogene-specific siRNAs targeting ros1 and egfr1 mRNA transcripts, than the strontium sulfite particles prepared in absence of NaCl and d-glucose, as confirmed by growth inhibition assay. The mouse plasma binding analysis by Q-TOF LC-MS/MS demonstrated less protein binding to smaller particles of Na-Glc-SSNs. The biodistribution studies of the particles after 4 h of treatment showed Na-Glc-SSNs had less off-target distribution than SSNs, and after 24 h, all siRNAs were cleared from all major organs except the tumors. ROS1 siRNA with its potential therapeutic role in treating 4T1-induced breast tumor was selected for subsequent in vivo tumor regression study, revealing that ROS1 siRNA-loaded SSNs exerted more significant anti-tumor effects than Na-Glc-SSNs carrying the same siRNA following intravenous administration, without any systemic toxicity. Thus, strontium sulfite emerged as a powerful siRNA delivery tool with potential applications in cancer gene therapy.

## 1. Introduction

Nanotechnology has contributed immensely to cancer gene therapy via introducing multiple nanocarriers to carry plasmid DNA, mRNA, siRNA, miRNA and anti-sense oligonucleotides (AONs) to the cytoplasm of targeted cancer cells [1]. Rigorous studies of cancer genetics have revealed that mutations in tumor suppressor genes and oncogenes are the key drivers for aberrant expression of genes, leading to cancer [2]. As a RNA interference tool, siRNAs have been deployed successfully for silencing genes that regulate proliferation, survival and metastasis of cancer cells [3]. However, unmodified or naked siRNA is prone to nuclease degradation, phagocytic clearance, interactions with serum proteins and renal clearance [4]. Moreover, siRNA also copes with the abnormal tumor microenvironment (TME) including the dense extracellular matrix (ECM), and leaky and heterogeneous vessels which contribute to increased interstitial fluid pressure (IFP) [5,6,7,8]. To overcome this hurdle, naked siRNA potentially requires a vehicle to be successfully delivered into the target cancer cells. Compared to the viral counterparts, non-viral vectors comprising both organic and inorganic nanoparticles (NPs) are more attractive for clinical use due to their superior advantages in terms of safety, scalability and drug-loading capability [9,10,11]. Inorganic NPs of gold, carbonate apatite, quantum dots, iron oxide, mesoporous silica, and so on, represent a highly sophisticated platform for delivering siRNAs into the target cancer cells and provide greater advantages depending on the type of cancers. These include unique physic-chemical properties, high biocompatibility, improved pharmacokinetics and pharmacodynamics properties and active intracellular delivery in contrast to organic ones [12,13,14,15]. NPs-based drug delivery systems responsive to acidic pH, hypoxia and hyperthermia have added a new dimension to cancer gene therapy, offering enhanced diffusion, increased cellular uptake and rapid intracellular drug release from cargo via the early endosomal escape mechanisms [16,17,18,19,20,21]. The relatively lower pH of tumor extracellular compartments along with intracellular acidic endosomes has been capitalized to develop NPs that trigger release of drugs before or after entering the tumor cells [22,23]. Few pH-sensitive organic NPs like micelles [24,25], liposomes [26], chitosan-silica nanospheres [27] and polymeric nanoparticles [28] have been utilized in facilitating tumor-specific release of several anticancer drugs following delivery in vivo. Although organic NPs exert tumor inhibitory effects in mice, manufacturing difficulties, poor biocompatibility and poor drug loading capabilities limit their overall success rate. Recently inorganic pH-sensitive carbonate apatite NPs were shown to significantly and specifically carry both small molecular anticancer drugs and nucleic acids to cancer cells [29,30,31,32,33]. 

ROS1 which belongs to a family of receptor tyrosin kinases (RTKs) plays a vital role in growth and differentiation of normal cells, and in the development and progression of different cancers. The chromosomal rearrangement in a variety of cancers activates the ROS1 proto-oncogene, causing cellular transformation. The aberrant expression of ROS1 is implicated in a variety of cancers including lung cancer [34] and breast cancers [35]. On the other hand, epidermal growth factor receptor (EGFR), which belongs to another family of RTKs lies at the head of a complex transduction cascade, modulating cell proliferation, survival, adhesion, migration and differentiation. With extracellular ligand binding domain, EGFR protects cancer cells against apoptosis and helps promote invasion and angiogenesis [36]. While growth-factor-induced EGFR signaling is essential for many normal morphogenic processes and numerous additional cellular responses, the aberrant activity of the receptor family has been shown to play a key role in the development and growth of tumor cells. Epidermal growth factor (EGF)-like proteins and neuregulins stimulate cells to divide by activating members of the EGFR family, which consists of the EGFR itself and the receptors known as HER2–4. It has been reported that expression level of EGFR is relatively high in TNBC (triple-negative breast cancer) with gene amplification in approximately 25% cases of TNBC [37,38]. Thus, silencing overexpressed ROS1 and EGFR could be a promising modality for breast cancer treatment.

Here we introduce strontium sulfite NPs (SSNs), which have an electrostatic affinity towards siRNAs and the ability to efficiently carry them into breast cancer cells. Strontium, an alkaline earth metal was found to exert pharmacological effects against osteoporosis and was used for radiotherapy for bone cancer treatment. Strontium carbonate NPs were used previously to carry the anticancer drug, etoposide into human gastric cancer cells [39]. In this article, we present an easy, simple and scalable method for synthesis of both SSNs and siRNA-SSNs complexes and demonstrate the roles of NaCl and glucose in synthesis and stabilization of Na-Glc-SSNs with more uniform size. Synthesized SSNs were characterized and their siRNA loading capability, cellular internalization, and cytotoxicity against MCF-7 breast cancer cells were evaluated. Finally, the potential of these particles in intracellular delivery of ros1- and egfr1-specific siRNAs was investigated through assessment of cytotoxicity in breast cancer cells, biodistribution study and tumor regression in a synergetic mouse model of breast cancer.

## 2. Materials and Methods

### 2.1. Materials 

Strontium chloride (SrCl_2_), sodium sulfite (Na_2_SO_3_), NaCl, d-glucose and 4-(2-hydroxyethyl)-1-piperazineethanesulfonic acid (HEPES) were obtained from Sigma-Aldrich (St. Louis, MO, USA). Dulbecco’s modified eagle medium (DMEM), dimethyl sulphoxide (DMSO), thiazolyl blue tetrazolium bromide (MTT), and ethylene diamine tetra acetic acid (EDTA) were purchased from Sigma-Aldrich (St. Louis, MO, USA). DMEM powder, fetal bovine serum (FBS), trypsin-ethylene diamine tetra acetate (trypsin-EDTA), and penicillin-streptomycin were from Gibco BRL (CA). All siRNAs used in this study were obtained from Qiagen (Hilden, Germany) and dissolved in RNase-free water provided by the company. MCF-7 cells were originally from ATCC (Manassas, VA, USA).

### 2.2. Cell Culture and Seeding

Human breast cancer cell line, MCF-7 was cultured in a 25 cm^2^ culture flask with complete DMEM (pH 7.4) supplemented with 10% heat-inactivated fetal bovine serum (FBS), 1% HEPES and 1% penicillin-streptomycin solution in a humidified atmosphere containing 5% CO_2_ at 37 °C. The cells from the exponential growth phase were trypsinized and seeded at 50,000 cells per well into a 24-well plate. 

### 2.3. Synthesis of SSNs and Na-Glc-SSNs

SSNs and Na-Glc-SSNs were prepared by adding 1–5 µL of a cation providing salt, SrCl_2_ (1 M) without or with 300 mM of NaCl and 200 mM of d-glucose into an anion providing salt, Na_2_SO_3_ (1 M) and incubating at 37 °C for 30 min. Subsequently, 10% FBS supplemented DMEM (pH 7.5) medium was added to top up to 1 mL of final particle suspension. Spectrophotometric analysis was done at 320 nm wavelength to measure turbidity of SSNs, Na-SSNs (prepared in the presence of NaCl), Glc-SSNs (prepared in the presence of glucose) and Na-Glc-SSNs (prepared in the presence of both NaCl and glucose) by using UV 1800 Spectrophotometer, Shimadzu, Japan. Microscopic observations of Na-Glc-SSN nanoparticles were made by using an Olympus Microscope CKX41. All experiments were conducted at room temperature and the data was plotted into a graph with mean ± SD. 

### 2.4. Particle Size, Zeta Potential Measurements and Observation of SSNs and Na-Glc-SSNs

Size of the NPs was measured by using Zeta sizer (Nano ZS, Malvern, Worcestershire, UK) in different SrCl_2_ concentrations with or without NaCl (300 mM) and glucose (200 mM). A refractive index (RI) of 1.325 was used to measure particle size and particle size distribution. The zeta potential of SSNs, Na-SSN, Glc-SSN and Na-Glc-SSN were also measured by using Zeta sizer (Nano ZS, Malvern, Worcestershire, UK). Zetasizer software 6.20 was employed to carry out the analysis of data and all samples were measured in triplicate. The date was plotted into a graph with mean ± SD. 

### 2.5. Characterization of Particles by a Field Emission Scanning Electron Microscope (FE-SEM) and an Energy Dispersive X-ray (EDX) Analyzer

The morphology of SSNs and Na-Glc-SSNs was observed by using FE-SEM. SSNs and Na-Glc-SSNs prepared by adding 60 mM of SrCl_2_, with or without 300 mM of NaCl, 200 mM of d-glucose to 10 mM of Na_2_SO_3_. 3 µL of particle suspension was then transferred to a glass slide for drying at 37 °C for 1 h. The dried sample was then placed onto a carbon tape-coated sample holder, followed by platinum sputtering of the dried samples with 30 mA sputter current at 2.30 tooling factor for 70 s and the sputtered particles were visualized at 5.00 kV using FE-SEM (Hitachi/SU8010, Tokyo, Japan) and analyzed for elemental composition by an EDX analyzer (X-maX, 50 mm^2^ HORIBA, Kyoto, Japan).

### 2.6. Characterization of NPs by Fourier Transform Infrared Spectroscopy (FT-IR) and X-ray Diffraction (XRD)

Na-Glc-SSNs and SSNs were prepared by addition of 60 mM of SrCl_2_, with or without 300 mM of NaCl, 200 mM of d-glucose to 10 mM of Na_2_SO_3_ and centrifuged at 3750 rpm for 30 min by using Allegra X-12 Centrifuge (Beckman Coulter, Fullerton, CA, USA), followed by removal of supernatant and re-centrifugation. Precipitated pellets were lyophilized by using a freeze dryer (Labconco, Kansas City, MO, USA). The spectrum of SSN and Na-Glc-SSN was observed by Varian FTIR using the Varian Resolution Pro 640 software (Agilent, Santa Clara, CA, USA) and XRD (inXitu, Mountain View, CA, USA). 

### 2.7. Acid Dissolution Study of SSNs and Na-Glc-SSNs

SSNs were prepared by mixing 60 mM of SrCl_2_ and 10 mM of Na_2_SO_3_ and Na-Glc-SSNs were prepared by the addition of 60 mM of SrCl_2_, 300 mM of NaCl, 200 mM of d-glucose and 10 mM of Na_2_SO_3_, followed by incubation for 30 min at 37 °C. After the incubation, DMEM of different pHs was added to make the final volume 1 mL. The different pHs were adjusted by using 1 N HCl and the absorbance was measured at 320 nm wavelength by using a spectrophotometer. Experiments were done in triplicate and the data were plotted with mean ± SD. 

### 2.8. Assessment of siRNA Binding Affinity to Particles

Different concentrations of AF 488 allstars negative siRNA were dissolved in 200 µL of DMEM and fluorescence intensity was measured with a λex = 490 nm and λem = 535 nm by using 2030 multilabel reader victor TM X5 (Perkin Elmer, MA, CA, USA). Data was analyzed by Perkin Elmer 2030 manager software. Each experiment was done in triplicate and a standard curve was obtained by plotting fluorescence intensity versus siRNA concentrations to calculate the amount of siRNA bound to the NPs. SSNs, Na-SSNs (prepared in the presence of NaCl), Glc-SSNs (prepared in the presence of Glucose) and Na-Glc-SSNs were prepared by addition of 60 mM of SrCl_2_, 300 mM of NaCl, 200 mM of d-glucose to 10 mM of Na_2_SO_3_ in presence of 10 nM of AF 488 siRNA and incubating for 30 min at 37 °C. After the samples were centrifuged at 13,000 rpm for 15 min at 4 °C, the supernatant was collected and fluorescence intensity measurement. The binding affinity of siRNA to differently formulated NPs was calculated by using the following formula:$$\%\;\mathrm{of}\;\mathrm{siRNA}\;\mathrm{binding}=\frac{X_{\mathrm{initial}}-X_{\mathrm{free}}}{X_{\mathrm{initial}}}\times 100\%$$
where *X*_free_ is the concentrations of siRNA in the supernatant following centrifugation of NPs-siRNA, and *X*_initial_ denotes the total concentration of siRNA used in the experiment, which was 10 nM. The samples were prepared in duplicate and represented as mean ± SD.

### 2.9. Cellular Uptake of siRNA-Loaded NPs 

MCF-7 cells were seeded in a 24-well plate (50,000 cells/well), incubated overnight and treated with the NPs formulated with 10 nM of AF 488 siRNA for observing cellular uptake of NPs-siRNA complexes. SSNs, Na-SSNs, Glc-SSNs and Na-Glc-SSNs were prepared as described above, through incubation for 30 min at 37 °C. After 4 h and 12 h of incubation, the treated cells were washed with 5 mM of EDTA in PBS for dissolving extracellular particles and observed under a fluorescence microscope (Olympus DP73, Tokyo, Japan).

Additionally, treated cells after 4 and 12 h were washed with PBS, treated with 5 mM EDTA in PBS and lysed before fluorescence intensity of the lysate was measured in 2030 multilabel reader victorTM X5 (Perkin Elmer) attached with Perkin Elmer 2030 manager software using an excitation wavelength of 490 nm and an emission wavelength of 535 nm. Samples were blank corrected using untreated samples. The experiment was completed in duplicate and expressed as mean ± SD.

### 2.10. Cell Viability Assay with MTT (3-(4,5-Dimethylthiazol-2-yl)-2,5-Diphenyltetrazolium Bromide)

Cytotoxicity of SSN, Na-SSN, Glc-SSN and Na-Glc-SSN in MCF-7 cells was assessed by MTT assay. 5 × 10^4^ cells were seeded in a 24-well plate in triplicate. On the following day, the cells were incubated with SSNs, Na-SSNs, Glc-SSNs and NA-Glc-SSNs for 48 h and MTT assay was conducted. Briefly, 50 µL of MTT (5 mg/mL in PBS) was added to each well and incubated for 4 h. After dissolving the resulting formazan products with 300 µL of DMSO (dimethyl sulfoxide), the absorbance was analyzed on a microplate reader (microplate spectrophotometer, Biorad, Hercules, CA, USA) at 595 nm wavelength with reference to 630 nm.

Cytotoxic effects of SSNs- and Na-Glc-SSNs-bound EGFR and ROS1 siRNAs in MCF-7 cells were also assessed by MTT assay. The siRNAs-loaded SSNs and Na-Glc-SSNs were prepared by adding 60 mM of SrCl_2_, without or with 300 mM of NaCl and 200 mM of d-glucose to 10 mM of Na_2_SO_3_ and 1 nM of the respective siRNA and incubating for 30 min at 37 °C. The particle suspension was finally topped up to 1 mL with 10% FBS-containing DMEM (pH 7.5). 5 × 10^4^ cells were seeded in a 24-well plate. On the following day, the cells were treated with SSNs, SSNs-EGFR siRNA, SSN-ROS1, Na-Glc-SSN, Na-Glc-SSN-EGFR and Na-Glc-SSN-ROS1, followed by incubation for 48 h, and finally, cytotoxicity was measured. Briefly, 50 µL of MTT (5 mg/mL in PBS) was added to each well prior to incubation of the plate for 4 h. After dissolving the resulting formazan crystals with 300 µL of DMSO (dimethyl sulfoxide), the absorbance was analyzed on a microplate reader (microplate spectrophotometer, Biorad, Hercules, CA, USA) at 595 nm wavelength with reference to 630 nm. The percent of metabolically active cells (CV) was calculated for treated samples using the following equation: $$\%\;\mathrm{cell}\;\mathrm{viability}=\frac{\mathrm{Absorbance}\;\mathrm{of}\;\mathrm{treated}\;\mathrm{sample}}{\mathrm{Absorbance}\;\mathrm{of}\;\mathrm{control}}\times 100\%$$

All the experiments were done in triplicate and the data was plotted as % of cell viability with mean ± SD. 

### 2.11. In-Solution Digestion of SSN and Na-Glc-SSN Protein Corona for Mass Spectrometric Analysis

SSNs and Na-Glc-SSNs were prepared by adding 60 mM of SrCl_2_, with or without 300 mM of NaCl, 200 mM of d-glucose to 10 mM of Na_2_SO_3_ and incubating for 30 min at 37 °C, and subjected to additional incubation with mouse plasma (10%) for 15 min at 37 °C. After centrifugation of the particle suspensions at 13,000 RPM for 15 min, the supernatants were discarded, and the pellets were washed in Milli Q water, followed by centrifugation and removal of the supernatants. The pellets were dissolved with 100 µL of 50 mM EDTA in H_2_O. 25 µL of 100 mM ammonium bicarbonate solution, 25 µL tetrafluoroethylene (TFE) denaturing agent and 1 µL of 200 mM dithiothreitol (DTT) solution were added to the protein mixture (released from pellets), followed by vortexing and heating under a heating block at 60 °C for 1 h. After adding 4 µL of 200 mM iodoacetamide (IAM) and briefly vortexing, the protein mixture (representing protein corona) was incubated in the dark at room temperature for 1 h. 1 µL of 200 mM DTT solution was added to the protein mixture which was then incubated in the dark at room temperature for 1 h. Afterwards, the treated protein mixture was incubated at room temperature in presence of 100 µL ammonium bicarbonate solution (100 mM) and MS Grade 25 µL of Trypsin (1 µg/mL) at 37 °C for 4 to 18 h. Finally, 1µl formic acid was added to stop the reaction, and the samples were subjected to speed vacuum overnight prior to analysis by Q-TOF LC-MS/MS.

### 2.12. Sample Preparation for Mass Spectrometry-Based Proteomics

10 µL of formic acid (0.1%) in water was added to dissolve dry peptide digest. Samples were then sonicated in ultrasonic water bath for 10 min, while maintaining room temperature using ice. Samples were centrifuged (14,000× *g*, 5 min) and 5 µL of supernatant was placed in MS tube before being directly transferred on LC-QTOF auto-sampler for analysis.

### 2.13. High Efficiency Nanoflow Liquid Chromatography Electrospray-Ionization Coupled with Mass Spectrometry

The peptides digested were loaded into an Agilent Poroshell 300 Å pore C18 columns (Agilent, Santa Clara, CA, USA) using 0.1% formic acid mobile phase to equilibrate the column. The peptides were eluted from the column with 90% acetonitrile in 0.1% formic acid (solution B), using the gradients of 5% solution B over 0–30 min and 75% solution B over 30–39 min. Quadrupole-time of flight (Q-TOF) polarity was set at positive with capillary and fragmented voltage being set at 1750 V and 360 V, respectively, and 5 L/min of gas flow with a temperature of 325 °C. The peptide spectrum was analyzed in auto MS mode ranging from 110–3000 m/z for MS scan and 50–3000 m/z for MS/MS scan. Acquisition rates were 2 (spectra/s) for MS and 4 (spectra/s) for MS/MS. The spectrum was then analyzed with Agilent MassHunter (Agilent Technologies, Santa Clara, CA, USA) data acquisition software and then PEAKS 8.0 software (Bioinformatics Solutions Inc., Waterloo, ON, Canada).

### 2.14. Protein Identification and Quantification by Automated De Novo Sequencing (PEAKS Studio 8.0)

Protein identification was performed by integrating a database search (SwissProt.Mus_musculus) with de novo sequencing, for the homology search using PEAKS Studio 8.0 (Bioinformatics Solution Inc., Waterloo, ON, Canada). Carbamidomethylation was set as the fixed modification with maximum mixed cleavages at 3. Parent mass and fragment mass error tolerance were both set 0.1 Da with monoisotopic mass as the precursor mass search type. Trypsin was selected as the enzyme for digestion. False discovery rate (FDR) of 1% and unique peptides ≥1 were used for filtering out inaccurate proteins. A-10lgP score of greater than 20 indicates that detected proteins are relatively high in confidence as it targets very few decoy matches above that threshold. Relative differential changes of proteins commonly found in different complex protein samples of SSNs were quantified using PEAKS Q protein quantification software. Label free quantification method is based on the relative intensities of peptide ion peak features detected in multiple samples. Feature detection is performed separately on each sample with more overlapped features, by using the EM (expectation-maximization) algorithm. The features of the same peptide from different samples are reliably aligned together using a high-performance retention time alignment algorithm. The groups are color-coded to be used in the heat map summary to distinguish the groups between two NPs and the intensity of a quantifiable peptide. The significance of a peptide is denoted by its −10LogP score. The cut off value was set at 20 which is equivalent to a *p*-value of 0.01. Heat Map displays the protein groups that passed the filters for quantitative analysis. The relative protein abundance is represented as a heat map of the representative proteins of each protein group. The representative proteins are clustered if they exhibit a similar expression trend across the samples. The hierarchical clustering is generated using a neighbor-joining algorithm with a Euclidean distance similarity measurement of the log2 ratios of the abundance of each sample relative to the average abundance. 

### 2.15. In Vivo Biodistribution Study of SSNs and Na-Glc-SSNs

For biodistribution study, female Balb/c mice (6–8 weeks old) of 20–25 g of body weights were obtained from the School of Medicine and Health Science Animal Facility, Monash University. The mice were maintained in 12:12 light:dark conditions and provided with ab libitum and water. All the experiments were done in accordance with the protocol approved by MONASH Animal Ethics Committee (MARP/2016/126). Approximately 1 × 10^5^ 4T1 cells (in 180 µL PBS) were injected subcutaneously on the mammary pad of mice (considered as day 1) and the mice were checked regularly for the outgrowth of tumor by touching the area of injection by index finger. The tumor bearing mice were administered with fluorescent AF-488 labeled neg. siRNA (75 mM) either in free or NPs-bound form through tail vein injection when the tumor volume reached 79 mm^3^. Mice were sacrificed humanely by cervical dislocation after 4 or 24 h of the administration. Afterwards, the heart, liver, kidney, spleen, lung, brain and tumor were collected and washed twice in chilled PBS, followed by addition of 500 µL lysis buffer per 500 mg of tissue mass. Tissues were lysed using a mechanical homogenizer with four strokes intermittently while maintaining the samples on ice till a completely homogenized solution was obtained. The solutions of tissue lysates were centrifuged for 25 min at 4 °C with 8000 rpm. 200 µL of the supernatant was added to each well of a 96-well opti-plate (Nunc) for measuring fluorescence intensity of AF-488 labeled siRNA with 2030 multilabel reader vitorTM X5 (Perkin Elmer) attached with Perkin Elmer 2030 manager software using λex = 490 nm and λem = 535 nm. Data were represented as mean ± SEM of fluorescence intensity/500 mg of tissue mass after the values were blank-corrected using an untreated group of mice for each tissue.

### 2.16. 4T1.-Induced Mouse Model of Breast Cancer and Anti-Tumor Activity of siRNA-Loaded NPs 

Female Balb/c mice (6–8 weeks old) with body weights of 20–25 g (obtained from the School of Medicine and Health Science Animal Facility, Monash University) were maintained in 12:12 light:dark conditions and provided with ab libitum and water. All the experiments were done in accordance with the protocol approved by MONASH Animal Ethics Committee (MARP/2016/126). Approximately 1 × 10^5^ 4T1 cells (in 100 µL PBS) were injected subcutaneously on the mammary pad. When the volume of the growing tumor reached an average of 39 mm^3^ at around Day 9–10, mice were grouped randomly with 4 mice per group and treated intravenously (tail-vein) at the right or left caudal vein, while the second dose was administered 3 days after the 1st dose. The length and width of tumor outgrowth were estimated using the Vernier caliper in mm scale over the period of 24 days, with the data subsequently presented as mean ± SEM of the tumor volumes of each group. The volume of the tumor was calculated based on the following formula:Tumor Volume (mm^3^) = ½ (length × width^2^).

### 2.17. Statistical Analysis 

Statistical analysis was done by using SPSS version 23 (Armonk, NY, USA). LSD post hoc test for one way ANOVA and independent-samples *t*-test were used for analyzing in vitro and in vivo data and comparing the significant difference. Data were considered statistically significant when * *p* < 0.05 and very significant when ** *p* < 0.001.

## 3. Results and Discussion

### 3.1. Generation of SSNs and Evaluating Effects of NaCl and Glucose on Regulation of Particle Growth

SSNs were synthesized by mixing different concentrations of SrCl_2_ (20–100 mM) and 10 mM of Na_2_SO_3_ and incubating at 37 °C for 30 min, followed by addition of 10% FBS-supplemented DMEM (pH 7.5) to make the final volume to 1 mL. The particles were characterized by UV-VIS spectrophotometer and optical image analysis. The increment of reactant concentration leads to higher rates of particle formation, and consequential self-aggregation to form bigger size particles. Since DMEM contains high concentrations of NaCl and glucose, it is necessary to identify their roles in particle growth and stabilization of particle size. Particles with diameters ranging from 20 to 200 nm are supposed to have excellent tumor accumulation capacity and higher circulation time [40,41,42,43]. To investigate the effects of NaCl and glucose on SSNs formation, we added 300 mM of NaCl and 200 mM of glucose to form Na-SSNs, Glc-SSNs and Na-Glc-SSNs. As shown in Figure 1A, particle formation was enhanced with increasing Sr^2+^ concentration, as represented by high turbidity. However, when we added 300 mM of NaCl to form Na-SSN particles, a significant drop in turbidity was observed for all Sr^2+^ concentrations. This could be achieved by minimizing charges via temporary ionic interaction between the SrSO_3_ and NaCl, thereby slowing down the rate of reaction for particle formation. On the other hand, addition of d-glucose also significantly decreased the growth of particles as demonstrated by lower turbidity in Figure 1A, suggesting that glucose could shield the particles, impeding the interactions between the particles for subsequent growth. However, addition of NaCl and glucose to form Na-Glc-SSNs demonstrated lower turbidity in comparison to either Na-SSNs or Glc-SSNs, indicating that the combination of NaCl and glucose could effectively reduce and stabilize particle growth by controlling particle aggregation. Optical microscopic visualization of Na-Glc-SSNs following incubation at 37 °C for 1 h in a 24-well plate showed an increasing number of aggregated particles with increasing Sr^2+^ concentrations (Figure 1B), in parallel with the increase in absorbance (Figure 1A).

Since particle diameter is crucial in regulating pharmacokinetics and tumor targeting of particle-loaded therapeutics, we measured the average particle size by Zetasizer and investigated the effects of NaCl and glucose in the stabilization of particle diameter. As shown in Figure 2, the average size of SSNs without NaCl and glucose ranged from 20 nm to 1.3 µm, with a trend of forming larger particle with increasing Sr^2+^ concentrations, while addition of NaCl and glucose to the reaction mixture reduced the particle sizes of SSNs (i.e., Na-Glc-SSNs), which varied from 20 nm to 463 nm depending on the concentration of SrCl_2_ in a similar trend as in the turbidity study (Figure 2A). The surface charge of NPs is crucial for stability in systemic circulation, cellular uptake and successful delivery of drug into the target tumor cytoplasm. The zeta potential of SSNs, Na-SSNs and Glc-SSNs were almost same and remained in the range of −10.7 mV to −11.9 mV, whereas incorporation of NaCl and glucose into the SSNs slightly reduced the zeta potential (−10.5). The combination of NaCl and glucose reduced the size and stabilized the growth of Na-Glc-SSN more significantly than SSNs formed without NaCl and glucose. We propose that NaCl prevents particle aggregation by temporary binding with SrSO_3_ particles through its cations (Na^+^) and anions (Cl^−^) (Figure 3A), whereas glucose may be partitioned between the particles, thereby producing less aggregated and small sized particles (Figure 3B). 

### 3.2. Characterization of SSNs and Na-Glc-SSNs by Zeta Sizer

Zeta Sizer was used to investigate the particle size distribution of SSNs and Na-Glc-SSNs. The PDI (poly dispersing index) value was measured to know the dispersion homogenicity of the particles. As shown in Figure 4A, the PDI value of SSN particles was 0.988, indicating the heterogenous distribution of SSNs. Particle size distribution by number and volume also demonstrated variably sized particles. On the other hand, Na-Glc-SSNs particles formed with 60 mM of SrCl_2_ and 10 mM of Na_2_SO_3_ in the presence of 300 mM of NaCl and 200 mM of glucose showed a PDI value of 0.290 (Figure 4B). The lower PDI value of Na-Glc-SSNs indicates that particle has less tendency to agglomerate and form smaller and uniformly distributed particles. Preclinical studies showed that the particles within a diameter range of 100–150 nm are able to access the liver and tumor tissues after IV administration [44]. As shown in Figure 4B, relatively small size particles in Na-Glc-SSNs constitute the majority of the particles, suggesting that siRNA-Na-Glc-SSN complexes would have favourable pharmacokinetics and greater ability to penetrate into target cells via endocytosis more efficiently than SSN-siRNA complex. 

### 3.3. Characterization of Differently Formulated Particles by FE-SEM

Field-emission scanning electron microscopy was used to observe the morphology and actual size of the prticles. As shown in Figure 5A, SSNs displayed a spherical shape with rough surface. The average diameter of the particles was in the range of 595–625 nm, suggesting formation of the large particle size due to self-aggregation. On the other hand, Na-Glc-SSNs showed relatively more particles of a spherical shape with relatively smaller size distribution (242–267 nm) (Figure 5B). The smaller particle size could be due to the effects of NaCl in stabilizing the particles by slowing down the reaction and d-glucose that lies between the particles in preventing self-aggregation.The smaller and rough surfaced SSNs were expected to demonstrate better drug binding ability as well as less protein binding affinity in systemic circulation.

### 3.4. Elemental Analysis of Nanoparticles by EDX

The EDX spectrum showed strong peaks of strontium around 2 KeV for both SSNs and Na-Glc-SSNs (Figure 6), confirming strontium as a major constituent. In addition, sulfur, oxygen and chlorine also showed peaks for both NPs with small amount of sodium, carbon, oxygen and platinum. The signals for sulfur and oxygen might originate from the sulfite group of the particles. Glucose and NaCl used in the fabrication of Na-Glc-SSNs contributed to the signals for carbon, oxygen, sodium and chloride. The other signals might come from platinum sputtering of the dried samples.

### 3.5. Characterization of SSNs and Na-Glc-SSNs by FT-IR and XRD 

The formation of strontium sulfite was confirmed via FT-IR, which involves the vibration of molecules targeted via the infrared spectroscopy on the lyophilized SSNs and Na-Glc-SSNs. The FT-IR spectra of glucose, SSNs and Na-Glc-SSNs were obtained. The spectra of SSNs displayed two strong peaks at 916 cm^−1^, 640 cm^−1^ and 521 cm^−1^, which are the characteristic peaks of ^•^Sulfite ion (SO_3_^2−^) (Figure 7). On the other hand, the similar peaks for SO_3_^2−^ in Na-Glc-SSNs were noted at around 913 cm^−1^, 631 cm^−1^ and 530 cm^−1^ with a small shift, which is probably due to the presence of strontium, NaCl and glucose. Moreover, the overall vibration pattern that is characteristic of SrSO_3_ strongly suggests that SrSO_3_ was successfully formed. The XRD peak profiles of SSNs and Na-Glc-SSN crystals were similar to the standard peak profile of SrSO_3_ (Figure 8). In addition, the peak broadening of the material could be indicative of its lower degree of crystallinity, an important factor determining particle dissolution rate.

### 3.6. Acid Dissolution Profiles of SSNs and Na-Glc-SSNs 

The success of nanoparticles-mediated siRNA therapy depends on the efficient release of siRNA from the carrier in order to silence the target mRNA in cytoplasm. NPs/siRNA complex after being internalized into the cell via endocytosis should escape from endosomes as well as avoid lysosomal degradation. Particles that can be dissolved in acidic endosomes are able to release the payload from the NPs. Particle dissolution could also lead to accumulation of ions (Ca^2+^, PO_4_^3−^ and CO_3_^2−^) with eventual development of osmotic pressure across the endosomal membrane, which might result in endosomal breakdown and release of the therapeutics in cytosol. As shown in Figure 9, the absorbance of both SSNs and Na-Glc-SSNs was decreased sharply with a decrease in pH and at pH 6, the particle were fully degraded with an absorbance value below 0.1 nm. The result suggests that SSNs and Na-Glc-SSNs were unstable and degraded in the acidic pH, thus indicating that the particles would be dissolved in the acidic environment of endosomes to facilitate early release of siRNA.

### 3.7. Assessment of Binding Affinity of siRNA to SSNs and Na-Glc-SSNs

The binding affinity of siRNA towards nanoparticles is vital to prevent its nuclease-mediated siRNA degradation and dissociation from the particles after being exposed to blood components. siRNA, which is negatively charged due to its phosphate backbone was expected to interact with strontium sulfite nanoparticles owing to their positively charged Sr^2+^-rich domains. As shown in Figure 10, SSNs, Na-SSNs, Glc-SSNs and Na-Glc-SSNs demonstrated significant binding affinity (ranging from 91% to 94%) towards the siRNA, suggesting that the potential ability of the nanoparticles in carrying siRNAs into cancer cells by preventing nuclease-mediated degradation and enhancing cellular uptake via endocytosis.

### 3.8. Cellular Uptake of Fluorescence-Labeled siRNA Carried by Differently Formulated Strontium Sulfite Nanoparticles

Cellular uptake of siRNA, one of the most critical steps in regulating overall silencing efficacy is predominantly influenced by siRNA binding affinity for nanoparticles and size of the particles. The small particles are expected to have more cellular internalization in comparison to larger ones. MCF-7 cells were treated with differently formulated strontium sulfite nanoparticles for 4 h prior to removal of extracellular particles with EDTA and subsequent observation by a fluorescence microscope. After 4 h of treatment, as shown in Figure 11A, untreated cells and cells treated with free siRNA did not show any fluorescence signals. On the other hand, cells treated with all different forms of strontium sulfite particles revealed significant fluorescence signal. However, after 12 h of treatment, cells treated with SSNs coupled siRNA gave more significant fluorescence signal as demonstrated in Figure 11B, suggesting their efficient internalization into the breast cancer cells in extended time. The fluorescence intensity of Na-Glc-SSNs at 4 h was much higher than that of free siRNA and SSNs, as shown in Figure 12A. 

On the other hand, the fluorescence intensity of Na-Glc-SSNs at 12 h were increased over an extended period of time (Figure 12B). The SSNs which were prepared in the absence of NaCl and glucose, with a much bigger size than the others, seemed to mainly adhere to the cell membrane rather than going into the cells, indicating that particle size dramatically influences the cellular uptake process.

### 3.9. Cell Viability Assessment with MTT Assay

The biocompatibility of strontium sulfite nanoparticles and the extent of toxicity were assayed through MTT colorimetric assay. As shown in Figure 13, the viability of differently fabricated strontium sulfite particles was approximately 80% compared to the untreated cells, indicating that the nanoparticles are apparently biocompatible. Around 20% of cell deaths could be due to sedimentation of too many particles on the cell surface in a culture system. 

### 3.10. Intracellular Delivery of EGFR and ROS1 siRNA Using SSNs and Na-Glc-SSNs

To investigate the efficiency of strontium sulfite as a nanocarrier to transport siRNA into the target tumor, we selected siRNAs to silence EGFR and ROS1 genes, which are essential in proliferating cancer cells. Epidermal growth factor (EGF) stimulates cells to divide by activating members of the EGF receptor (EGFR) family which are the members of receptor tyrosine kinases that modulate cell proliferation, survival, adhesion, migration and differentiation. Although growth-factor-induced EGFR signaling is essential for many normal morphogenic processes and cellular responses, the aberrant expression of EGFR genes in different tumor cells has been noted [45]. The product of proto-oncogene ROS1 is another member of receptor tyrosine kinases (RTKs) that regulates the initiation and progression of various types of cancers. Several chromosomal rearrangements and aberrant expression of ROS1 gene were found in acute lymhoplastic leukemia, ovarian cancer, malignant gliomas, non-small cell lung cancer and breast cancer cells [30,46,47,48,49,50]. Silencing of EGFR or ROS1 gene expression could be a potential treatment modality for breast cancer through suppressing cancer aggressiveness. As shown in Figure 14, while SSNs with loaded EGFR or ROS1 siRNA did not show statistically significant toxicity in comparison to SSNs treatment, Na-Glc-SSNs with the same siRNAs showed significant cytotoxicity in MCF-7 cells in comparison to Na-Glc-SSNs treatment. The high variations in viability of the cells treated with SSNs could be due to their inconsistent, strong associating with cell membrane. On the other hand, Na-Glc-SSNs, which are quite small in size showed a promising outcome following delivery of the therapeutic siRNAs.

### 3.11. Analysis of Protein Corona Formed onto SSNs and Na-Glc-SSNs

The interactions between the proteins present in the serum and the surface of SSNs were analyzed by LC-MSLC-MS/MS Q-TOP. The in-solution digestion of the protein corona formed around SSNs and Na-Glc-SSNs was carried out to detect the peptides. The peptides derived from de novo sequencing were identified as exact or homologous peptides using the Mus_musculus database (SwissProt). The protein corona profile was characterized with the help of unique peptide, molecular weight, coverage % for peptides and significance (−10lgp). Detected proteins were listed along with their functions in Table 1 and Table 2 for SSNs and Na-Glc-SSNs. Protein classification based on their biological functions was plotted in a pie chart (Figure 15).

As shown in Figure 15, SSNs prepared without NaCl and glucose possessed affinity for different types of proteins including structural proteins (different keratins, Nup205), transport proteins (albumin, oligomeric Golgi complex subunit 7), and enzymes (protein kinases, endonucleases, glutamine synthetase), whereas SSNs prepared in the presence of glucose and NaCl showed affinity for more selective proteins, such as structural proteins (keratins) and enzymes (helicase), which could be due to the effects of Na^+^ and Cl^-^ ions in interfering in the weak ionic interactions of proteins with the particles. Serum albumin, the most abundant protein in the blood and a dysopsonin were detected in SSNs.

### 3.12. In Vivo Biodistribution Study of SSNs and Na-Glc-SSNs

The biodistribution and tumor accumulation of siRNA was investigated upon intravenous injection of AF-488-siRNA-complexed nanoparticles into mice bearing a subcutaneous 4T1 tumor. After 4 h of the treatment, major organs and tumor were collected and fluorescence intensity was measured. We observed significant differences in the biodistribution and tumor accumulation of the siRNA electrostatically associated with SSNs and Na-Glc-SSNs, as shown in Figure 16. The SSNs were significantly accumulated in RES organs (liver, spleen), brain, respiratory systems, heart and urinary systems. On the contrary, accumulation of Na-Glc-SSNs was lesser in major organs than that of SSNs except kidneys, which showed higher fluorescence signals, which could be explained by the notion that smaller particle size distribution of the former led to accelerated renal clearance. This might be reversed by increasing the hydrodynamic diameter via coating Na-Glc-SSNs with a highly polymer like PEG, etc. After 24 h of the treatment, both nanoparticles were completely eliminated from the body (Figure 16). The tumor accumulation of nanoparticles depends on their physicochemical properties to cross the TME. The EPR effects and vascular pore cut off size ranging from 200 nm to 1.2 um of tumors allow the nanoparticles to accumulate in the tumor region more efficiently [51]. We found that the SSNs and Na-Glc-SSNs facilitated significant tumor accumulation of the loaded siRNA after 4 h of the treatment and remained after 24 h of the treatment. Interestingly, there was a noticeable siRNA accumulation in the brain at 4 h, followed by complete removal after 24 h. 

### 3.13. In Vivo Anti-Tumor Effect of ROS1 siRNA-Loaded SSNs and Na-Glc-SSNs 

A 4T1-induced murine breast cancer model was used to investigate the effect of SSNs and Na-Glc-SSNs carrying ROS1 siRNA on tumor regression. 50 µL volumes of SSNs-ROS1 and Na-Glc-SSNs-ROS1 prepared with 60 mM SrCl_2_, 10 mM of Na_2_SO_3_ and 50 nM of ROS1 siRNA in the absence and presence of 300 mM of NaCl and 200 mm of d-glucose, respectively, were administered twice within an interval of 3 days through the tail vein, after the tumour volume reached 39 mm^3^. The results of the tumor growth curve (Figure 17A) show that tumors, following intravenous injections of SSNs and Na-Glc-SSNs, prepared under the same conditions except addition of siRNA, grew quickly almost in the same pattern. In contrast, the tumor growth of the mice treated with SSNs-ROS1 or Na-Glc-SSNs-ROS1 exhibited a relatively slower rate and smaller tumor volumes at the end, suggesting that both SSNs-ROS1 and Na-Glc-SSN-ROS1 effectively reduced the tumor growth in vivo. However, the tumor growth inhibition rate for SSNs-ROS1 was more prominent than Na-Glc-SSN-ROS1 at the later stage, which could be explained by higher renal clearance and lesser tumor accumulation of the latter. None of the two different particle types showed any obvious influence on mouse body weight (Figure 17C), which suggests that siRNA-loaded particles do not have any detrimental systemic toxicity in mice. The notable anti-tumor effects of SSNs and Na-Glc-SSNs carrying ROS1 siRNA indicates that strontium sulfite particles could protect the loaded siRNA against nuclease attack, confer favourable pharmacokinetics, induce no noticeable systemic cytotoxicity, promote excellent tumor uptake and cytosolic release of the siRNA. 

Since tumor extracellular environment is acidic, dissolution of the pH-sensitive particles could not be ruled out, with the consequence of siRNA release from the particles before their internalization into the target cells. We assume that under that acidic environment, particles could be dissolved either partially or completely, depending on the number of particles accumulated there. If the particles are partially dissolved, siRNAs released thereby might enter the target tumor cells by passive diffusion, which is rather inefficient considering the size of siRNA and the electrostatic repulsion between the negatively charged phosphate backbone of siRNA and the cell membrane, which is also negatively charged due to the presence of sulfate-carrying proteoglycans while the intact particles with electrostatically associated siRNA would enter via endocytosis resulting in particle dissolution under endosomal acidic pH and siRNA release from endosomes to cytosol. On the other hand, if the particles are fully dissolved in the extracellular compartment, the released siRNA would enter the tumor cells exclusively via passive diffusion with the consequence of poor therapeutic outcome. Since we have observed robust anti-tumor effect, we presume that the particles were partially dissolved or remained resistant to degradation under the mildly acidic environment.

## 4. Conclusions

SSNs and Na-Glc-SSNs synthesized via a simple precipitation method emerge as pH-sensitive smart carriers for effectively delivering therapeutic siRNAs to the tumor. We found significant roles of NaCl and d-glucose in stabilizing the strontium sulfite particles with a more uniform particle size of 220 nm. The low pH-triggered dissolution of strontium sulfite apparently helps in the efficient release of the siRNA in early endosomes, which is critical for silencing the target mRNA in cytosol. We have also found a significant effect of SSNs coupled siRNA in reducing tumor both in vitro and in vivo without any apparent toxicity. To our knowledge, this is the first successful report revealing that strontium sulfite nanoparticles are highly promising siRNA delivery vehicles to facilitate target oncogene knockdown and promote efficient tumor regression. Our findings therefore emphasize the need to perform further studies with strontium sulfite for tumor-targeted delivery of therapeutics in preclinical cancer models for potential translation into clinics in the future. 

## Figures and Tables

**Figure 1 pharmaceutics-11-00089-f001:**
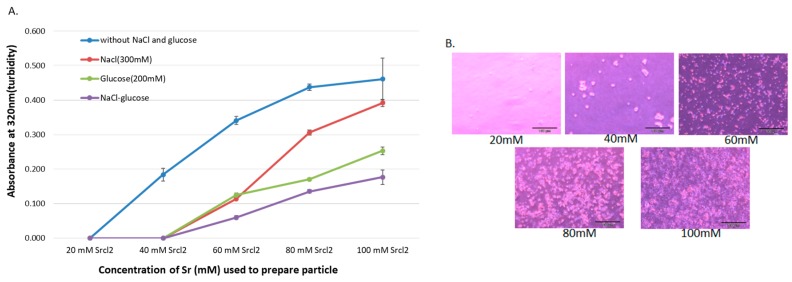
(**A**). Turbidity and optical microscopic analysis of differently formulated NPs. Turbidity (absorbance at 320 nm) of the particles formed without and with NaCl and/or glucose. Particles were formed by addition of 1–5 μL of (1 M) SrCl_2_ to 0.5 μL of (1 M) Na_2_SO_3_ in 50 µL aqueous solution with or without inclusion of 25 μL of NaCl (0.5 M) and/or 10 μL of d-glucose (1 M), and incubation for 30 min at 37 °C. Subsequently, serum-supplemented DMEM media was added to achieve 1 mL particle suspension. Absorbance at 320 nm wavelength was measured for all generated NPs using a spectrophotometer. (**B**). Microscopic observation of Na-Glc-SSNs prepared at different Sr^2+^ concentrations, and images were captured at 10× resolution.

**Figure 2 pharmaceutics-11-00089-f002:**
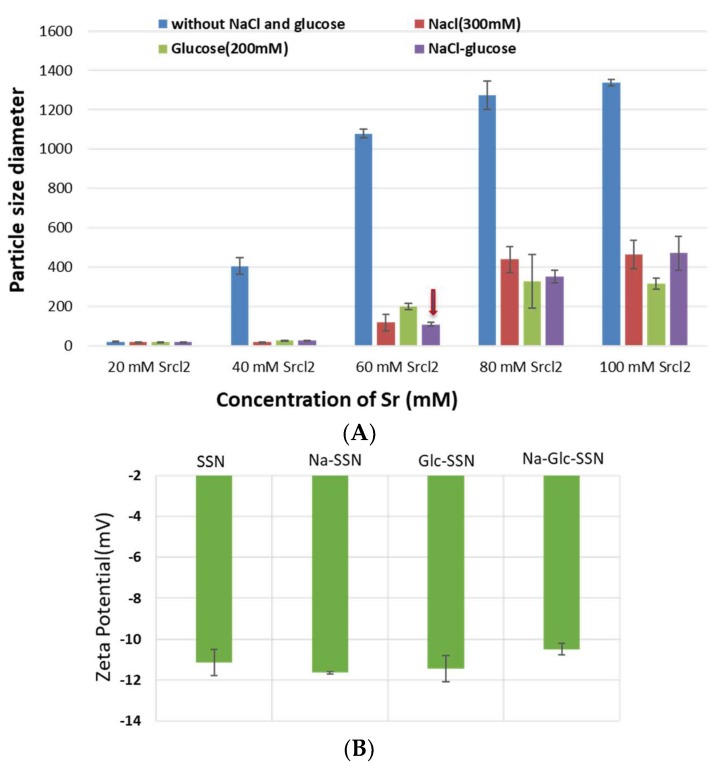
Size (**A**) and zeta potential (**B**) of different strontium sulfite nanoparticles with or without NaCl and glucose. Strontium sulfite nanoparticles were prepared by adding 60 mM of SrCl_2_, 300 mM of NaCl, 200 mM of glucose and fixed amount of Na_2_SO_3_.

**Figure 3 pharmaceutics-11-00089-f003:**
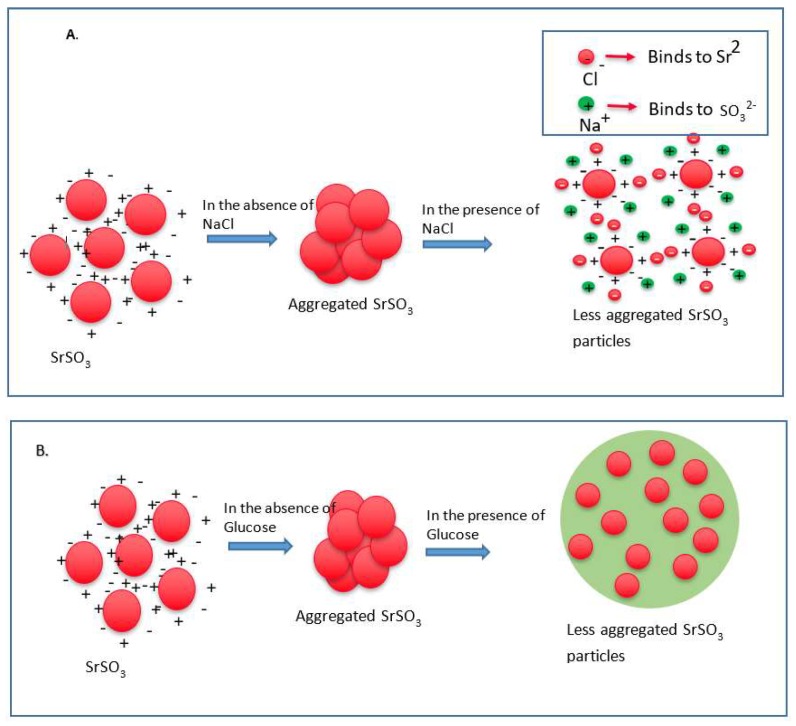
Schematic representation of effects of NaCl and glucose in stabilization of SrSO_3_ particles. (**A**) SrSO_3_ particles stick to each other to form aggregates, while NaCl slows down the particle formation by temporally binding with Sr^2+^ and SO_3_^2−^. (**B**) Without d-glucose, SrSO_3_ particles bind to each other forming large particles, but glucose lies between the particles and forms a shield to prevent particle-particle agglomeration.

**Figure 4 pharmaceutics-11-00089-f004:**
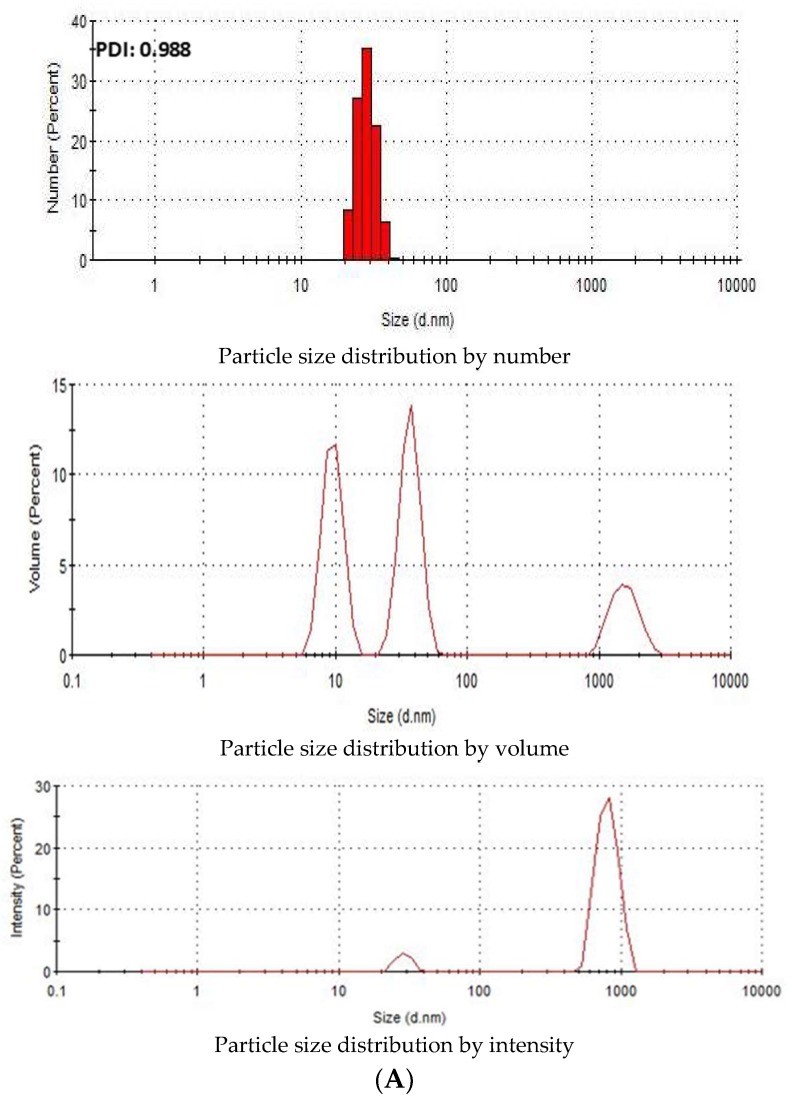
Particle size distribution of (**A**). SSNs and (**B**). Na-Glc-SSNs. SSNs and Na-Glc-SSNs were prepared by adding 60 mM of SrCl_2_, 300 mM of NaCl, 200 mM of glucose and 10 mM of Na_2_SO_3_.

**Figure 5 pharmaceutics-11-00089-f005:**
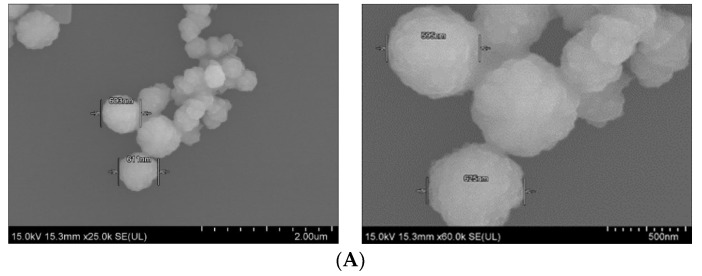
FE-SEM images of nanoparticles. (**A**). Micrographs of SSNs at 2 µm and 500 nm scale, (**B**). Micrographs of Na-Glc-SSNs at 5 µm and 3 µm scale.

**Figure 6 pharmaceutics-11-00089-f006:**
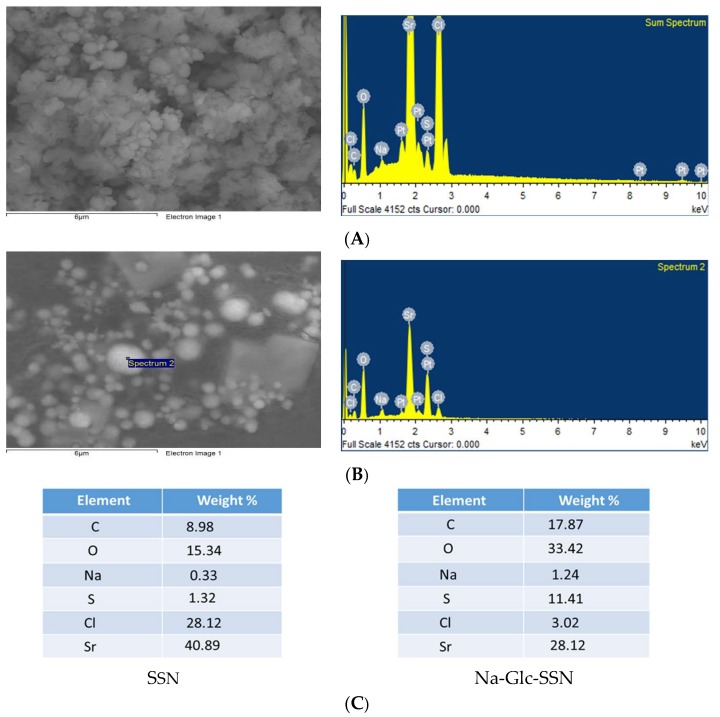
EDX analysis of nanoparticles. EDX spectrum of SSNs (**A**) and Na-Glc-SSNs (**B**) nanoparticle (**C**) Element weight percent of SSNs and Na-Glc-SSNs.

**Figure 7 pharmaceutics-11-00089-f007:**
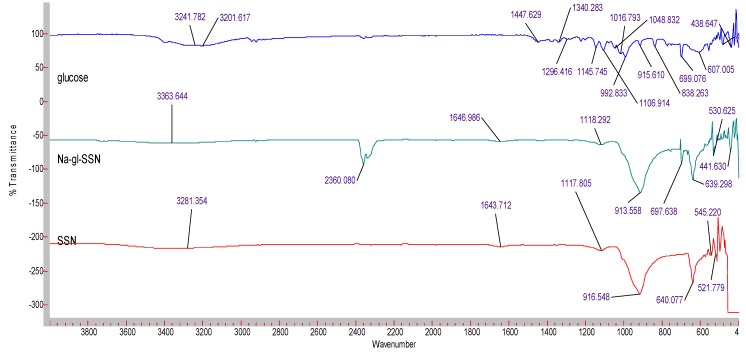
FTIR spectra for free glucose, SSNs (formed using 60 mM of SrCl_2_ and and 10 mM of Na_2_SO_3_) and Na-Glc-SSNs (formed by adding 60 mM of SrCl_2_, 300 mM of NaCl, 200 mM of glucose and 10 mM of Na_2_SO_3_).

**Figure 8 pharmaceutics-11-00089-f008:**
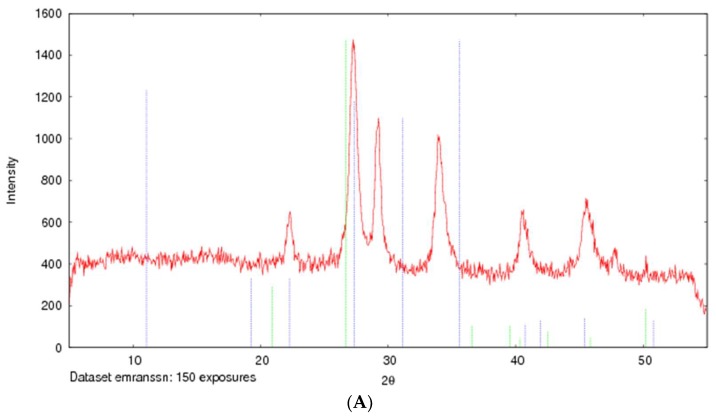
XRD patterns of SSNs (formed with 60 mM of SrCl_2_ and 10 mM Na_2_SO_3_, (**A**) and Na-Glc-SSNs (formed with 60 mM of SrCl_2_, 300 mM of NaCl, 200 mM of glucose and 10 mM of Na_2_SO_3_) (**B**).

**Figure 9 pharmaceutics-11-00089-f009:**
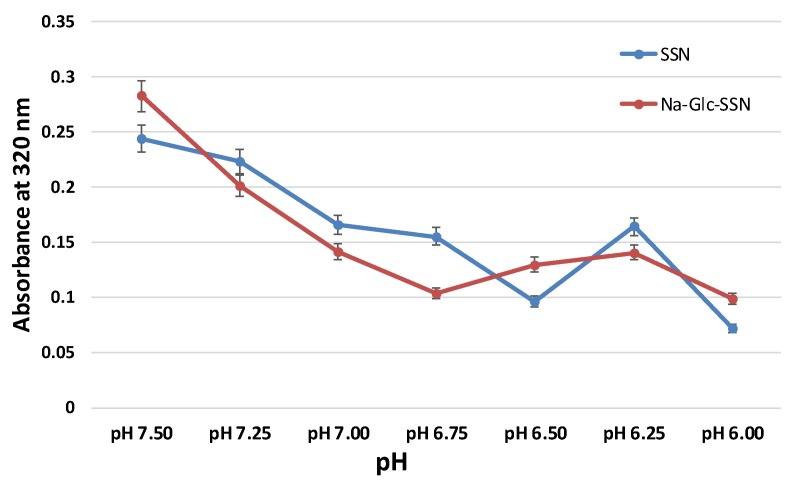
Acid dissolution profiles of SSNs, prepared by adding SrCl_2_ (60 mM) and Na_2_SO_3_ (10 mM) and Na-Glc-SSNs, prepared by adding 60 mM of SrCl_2_, 300 mM of NaCl, 200 mM of glucose and 10 mM of Na_2_SO_3_, in a 50 µL of an aqueous solution.

**Figure 10 pharmaceutics-11-00089-f010:**
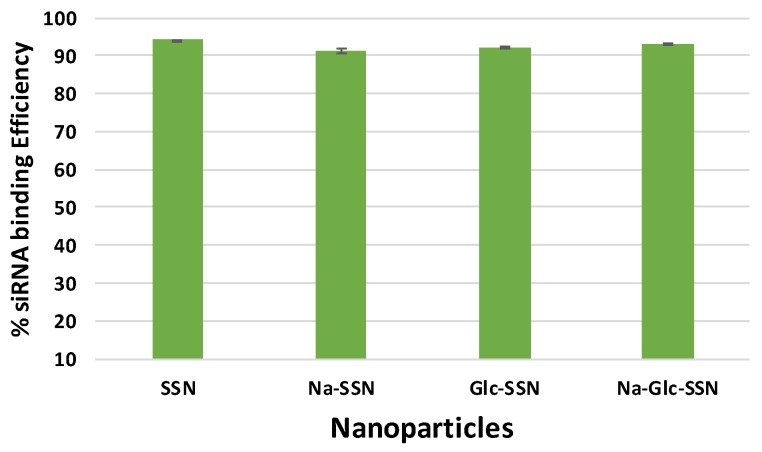
Binding efficiency of negative siRNA with differently formulated strontium sulfite particles formed by mixing SrCl_2_ (60 mM) and Na_2_SO_3_ (10 mM) along with 10 nM AF488 negative siRNA in the presence or absence of NaCl (300 mM) and glucose (200 mM).

**Figure 11 pharmaceutics-11-00089-f011:**
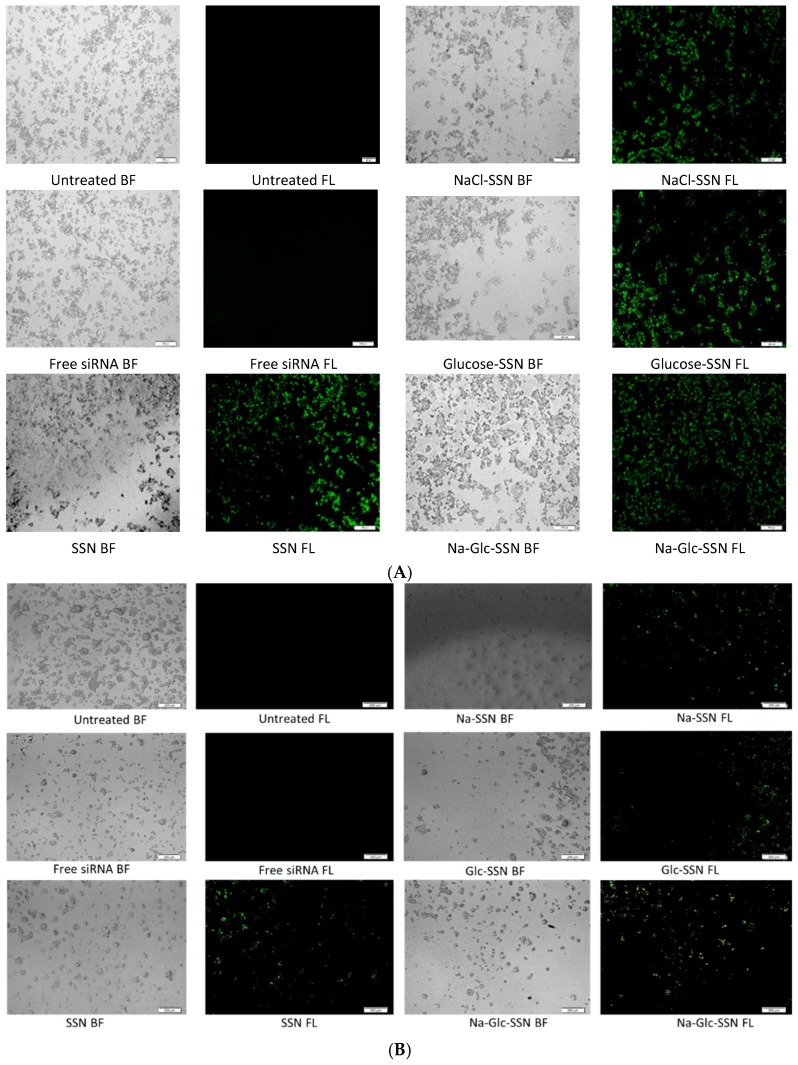
Cellular uptake of strontium sulfite particles with bound fluorescence-labeled siRNA. MCF-7 cells were treated with media (untreated), free siRNA and different strontium sulfite particles. SSNs, NaCl-SSNs, Glc-SSNs and Na-Glc-SSNs with loaded siRNA were formed by adding SrCl_2_ (60 mM), Na_2_SO_3_ (10 mM) and 10 nM of AF488 negative siRNA in presence or absence of NaCl (300 mM) and/or glucose (200 mM) at (**A**) 4 h of treatment and (**B**) 12 h of treatment.

**Figure 12 pharmaceutics-11-00089-f012:**
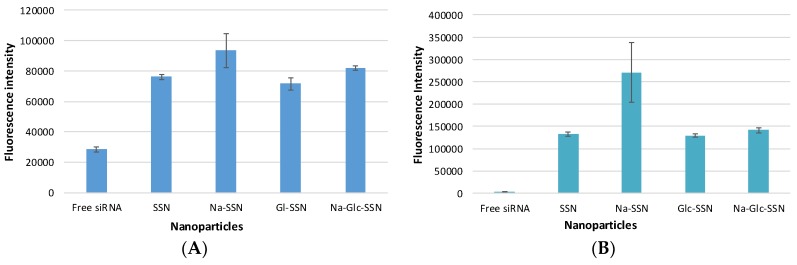
Fluorescence intensity of cell lysates. MCF-7 cells were treated with free siRNA, and NPs-siRNA formed with 60 mM of SrCl_2_ and 10 nM AF-488 negative control siRNA. After (**A**) 4 h and (**B**) 12 h, fluorescence intensity of cell lysates was measured. Values are representative of duplicate samples.

**Figure 13 pharmaceutics-11-00089-f013:**
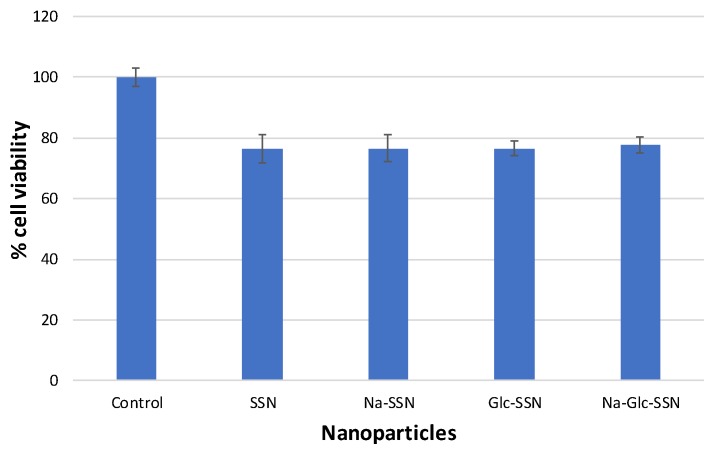
Cell viability assessment of different strontium sulfite particles formed with or without NaCl and d-glucose. MCF-7 cells were seeded and treated with SSNs (60 mM SrCl_2_ and 10 mM of Na_2_SO_3_, Na-SSN (60 mM SrCl_2_, 10 mM of Na_2_SO_3_ and 300 mM of NaCl), Glc-SSN (60 mM SrCl_2_, 10 mM of Na_2_SO_3_ and 200 mm of d-glucose) and Na-Glc-SSN (60 mM SrCl_2_, 10 mM of Na_2_SO_3_, 300 mM of NaCl and 200 mm of d-glucose).

**Figure 14 pharmaceutics-11-00089-f014:**
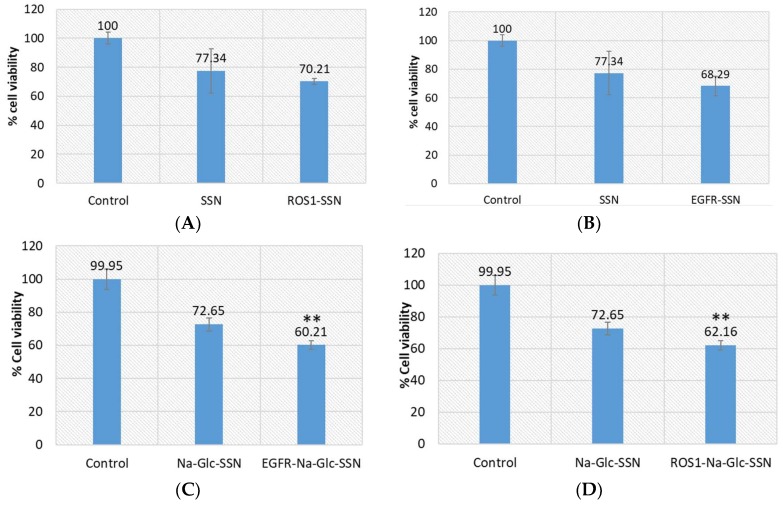
Cell viability assay following exposure of MCF-7 cells to SSNs and Na-Glc-SSNs with loaded ROS1 or EGFR siRNA. Cells were treated with particles alone and siRNA-loaded particles. SSNs and Na-Glc-SSNs were prepared by adding 60 mM SrCl_2_, without and with 300 mM of NaCl and 200 mm of d-glucose into 10 mM of Na_2_SO_3_. 1 nM of siRNA was used for fabrication of siRNA-loaded particles. Values are represented as % of cell viability in comparison to control for triplicate samples. (**A**). SSNs with loaded siRNA against ROS1. (**B**). SSNs with loaded siRNA against EGFR. (**C**). Na-Glc-SSNs with loaded siRNA against ROS1. (**D**). Na-Glc-SSNs with loaded siRNA against EGFR. Values were very significant (**) at *p* value 0.001 to 0.01 compared to NPs treatment.

**Figure 15 pharmaceutics-11-00089-f015:**
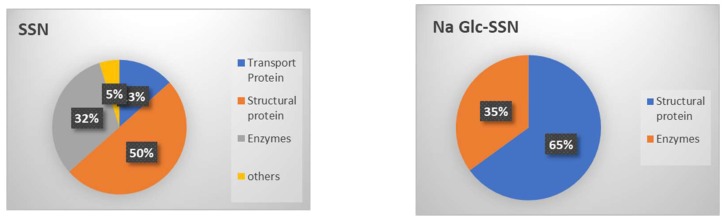
Protein corona profiling after 30 min incubation of SSNs (**left**), and Na-Glc-SSNs (**right**) with 10% of mice plasma. The detected proteins were classified based on their biological functions.

**Figure 16 pharmaceutics-11-00089-f016:**
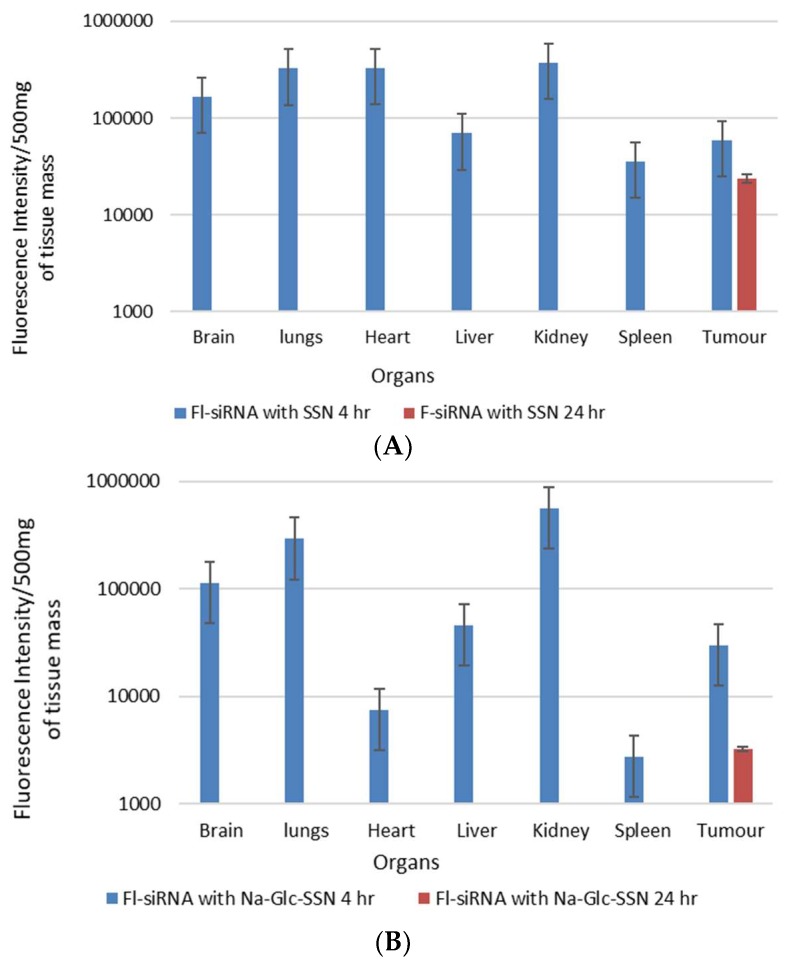
In vivo biodistribution of AF-488–labelled siRNA in nude mice bearing a xenografted 4T1 tumor 4 h and 24 h after intravenous injection of siRNA-loaded nanoparticles; (**A**). SSNs and (**B**). Na-Glc-SSNs. Each bar represents the mean ± SEM based on three mice per groups in all cases.

**Figure 17 pharmaceutics-11-00089-f017:**
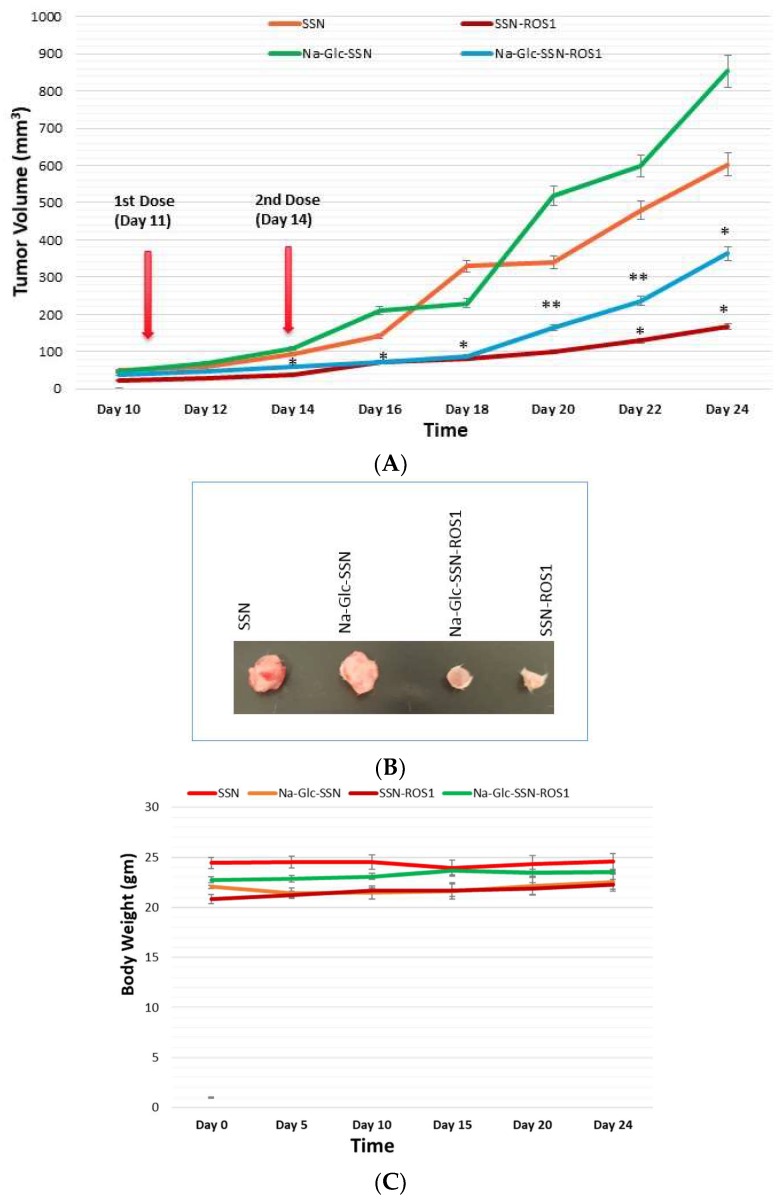
Effects of intravenously administered nanoparticles and siRNA-loaded nanoparticles on tumor regression. (**A**), Images of excised tumors at day 20 (**B**) and total body weight (**C**) in a 4T1 tumor bearing mice. 4T1 cells were inoculated subcutaneously on the mammary pad of mice. Tumor-bearing mice were treated intravenously through tail vein injection with 50 µL of either nanoparticles or siRNA-loaded nanoparticles, formed by the addition of 60 mM SrCl_2_, 300 mM of NaCl, and 200 mm of d-glucose to 10 mM of Na_2_SO_3_ in the absence and presence of 50 nM of ROS1 siRNA, respectively, when the tumor volume reached approximately 39 mm^3^. Four mice per group were used and data were represented as mean ± SEM of tumor volume. Values were very significant (**) at *p* value 0.001 to 0.01 and significant (*) at *p* value 0.01 to 0.05 as compared to the control group.

**Table 1 pharmaceutics-11-00089-t001:** Proteins bound with SSNs in the presence of 10% of mice plasma.

Protein Classes	Identified Proteins	−10lgP	Coverage	Mass	Function
Transport proteins	Albumin 1	156.86	31	68,693	chaperone binding, DNA binding, fatty acid binding, identical protein binding, oxygen binding, pyridoxal phosphate binding and toxic substance binding
Enzymes	Glutamine synthetase	38.65	4	42,019	glutamine biosynthetic process
Structural proteins	Keratin 16	32.67	1	51,606	structural constituent of cytoskeleton
Structural proteins	Keratin 16	32.67	1	51,693	structural molecule activity
Structural proteins	Keratin intermediate filament 16b	32.67	1	51,966	structural molecule activity
Structural proteins	Keratin intermediate filament 16a	32.67	1	52,053	structural molecule activity
Structural proteins	Uncharacterized protein	24.55	2	33,882	structural molecule activity
Structural proteins	Keratin 24 variant 2	24.55	2	40,994	structural molecule activity
Structural proteins	Keratin 19	24.55	2	44,542	protein-containing complex binding, structural constituent of muscle
Structural proteins	Keratin, type I cuticular Ha2	24.55	2	51,153	structural molecule activity
Structural proteins	Keratin 15, isoform CRA_a	24.55	2	49,494	scaffold protein binding, structural molecule activity
Structural proteins	Keratin, type I cytoskeletal 10	24.55	1	57,041	protein heterodimerization activity, structural constituent of epidermis
Structural proteins	Nup205	23.83	1	69,494	structural constituent of nuclear pore
Transport Proteins	Conserved oligomeric Golgi complex subunit 7	23.83	1	80,582	intracellular protein transport
Transport Proteins	Conserved oligomeric Golgi complex subunit 7	23.83	1	86,075	intracellular protein transport
Enzymes	Ercc5 protein	22.76	1	86,901	endonuclease activity, single-stranded DNA binding
Enzymes	Nek1 protein	21.82	2	48,636	ATP binding, protein serine/threonine kinase activity
Enzymes	Nek1 protein	21.82	1	133,856	ATP binding, protein kinase activity
Enzymes	Nek1 protein	21.82	1	139,659	ATP binding, protein kinase activity
Enzymes	MKIAA1901 protein	21.82	1	139,947	ATP binding, protein kinase activity
Enzymes	Nek1 protein	21.82	1	144,269	ATP binding, protein kinase activity
others	WD repeat-containing protein 81	21.82	1	211,931	mitochondrion organization

**Table 2 pharmaceutics-11-00089-t002:** Proteins bound with Na-Glc-SSNs in the presence of 10% of mice plasma.

Protein Classes	Identified Proteins	−10lgP	Coverage (%)	Mass	Functions
Structural proteins	Keratin, type I cytoskeletal 10	127.77	22	57,041	Protein heterodimerization activity, structural constituent of epidermis.
Structural proteins	Keratin, type II cytoskeletal 6B	112.44	9	59,526	structural molecule activity
Structural proteins	Krt6b protein	112.44	9	60,191	structural molecule activity
Structural proteins	Krt6b protein	112.44	9	60,273	structural molecule activity
Structural proteins	Keratin 77	106.07	6	61,302	structural molecule activity
Structural proteins	Keratin 77	106.07	6	61,359	structural molecule activity
Structural proteins	Keratin Kb40	69.48	2	85,239	structural molecule activity
Structural proteins	Keratin 78	69.48	1	112,265	structural molecule activity
Structural proteins	Type II cytokeratin Kb40	69.03	3	47,619	structural molecule activity
Structural proteins	Krt78 protein	63.78	3	54,765	structural molecule activity
Structural proteins	Krt78 protein	63.78	3	56,780	structural molecule activity
Structural proteins	Krt78 protein	63.78	3	54,774	structural molecule activity
Structural proteins	Keratin 15, isoform CRA_a	68.25	4	49,494	scaffold protein binding, structural molecule activity
Structural proteins	Uncharacterized protein	47.85	2	58,266	structural molecule activity
Structural proteins	Uncharacterized protein	47.85	2	58,240	structural molecule activity
Structural proteins	Keratin 90	47.85	2	58,224	structural molecule activity
Structural proteins	Krt2 protein	42.78	1	70,923	structural molecule activity
Enzymes	Eif4a1	29.17	3	33,069	ATP binding, ATP-dependent RNA helicase activity, translation initiation factor activity
Enzymes	Eukaryotic initiation factor 4A-II	29.17	3	36,166	ATP binding, ATP-dependent RNA helicase activity, translation initiation factor activity
Enzymes	Eukaryotic initiation factor 4A-II	29.17	2	41,290	ATP binding, ATP-dependent RNA helicase activity, translation initiation factor activity
Enzymes	Eif4a1	29.17	2	41,491	ATP binding, ATP-dependent RNA helicase activity, translation initiation factor activity
Enzymes	Eif4a1 protein	29.17	2	46,023	ATP binding, ATP-dependent RNA helicase activity, translation initiation factor activity
Enzymes	Eif4a1	29.17	2	46,184	ATP binding, ATP-dependent RNA helicase activity, translation initiation factor activity
Enzymes	Eif4a1	29.17	2	46,140	ATP binding, ATP-dependent RNA helicase activity, translation initiation factor activity
Enzymes	Eif4a1	29.17	2	46,154	ATP binding, ATP-dependent RNA helicase activity, translation initiation factor activity
Enzymes	Eukaryotic translation initiation factor 4A2	29.17	2	46,402	ATP binding, ATP-dependent RNA helicase activity, translation initiation factor activity

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
