# Peer review of "Strontium Sulfite: A New pH-Responsive Inorganic Nanocarrier to Deliver Therapeutic siRNAs to Cancer Cells"

_pharmaceutics, 2019, doi:10.3390/pharmaceutics11020089_

Round 1
Reviewer 1 Report
In this manuscript, the author synthesized strontium sulfite nanoparticle (SSN) and Na-Glc coated strontium sulfite nanoparticle (Na-Glc-SSN) as the nanocarriers for the efficient delivery of siRNA for the purpose of treating breast cancer. The design is interesting and the study was well performed. However, a few issues still need to be clarified prior to the further consideration of this manuscript.
1. In the animal assay, the final conclusion is that “the tumor growth inhibition rate for SSNs-ROS1 was more prominent than Na-Glc-SSN-ROS1 at the later stage”, then contradicting the advantages of Na-Glc-SSN in the study.
2. The description in the abstract is not clear enough, please make deep discussion.
4. In the section “Cellular uptake of fluorescence-labeled siRNA”, MCF-7 cells were only treated with siRNA nanoparticle for 4 hours. Please extend the time to compare the uptake of siRNA.
5. In the Fig. 11(A), “cells treated with free siRNA did not show any fluorescence signals”, please give a reasonable explanation.
6. Please add images of the biodistribution in vivo.
7. Most of the image pixels in this manuscript are too low to be identified. Please make further modifications.
8. The grammar of the manuscript needs to be modified.
9. Please make sure the punctuation and abbreviation are accurate and proper.
10. Some references are outdated. The reference lists should be updated with the recently published studies in the latest three years.
Author Response
Reviewer #1:
Comments and Suggestions for Authors
In this manuscript, the author synthesized strontium sulfite nanoparticle (SSN) and Na-Glc coated strontium sulfite nanoparticle (Na-Glc-SSN) as the nanocarriers for the efficient delivery of siRNA for the purpose of treating breast cancer. The design is interesting and the study was well performed. However, a few issues still need to be clarified prior to the further consideration of this manuscript.
1. In the animal assay, the final conclusion is that “the tumor growth inhibition rate for SSNs-ROS1 was more prominent than Na-Glc-SSN-ROS1 at the later stage”, then contradicting the advantages of Na-Glc-SSN in the study.
Response: In tumor regression study using a syngeneic mouse model, we got 78% and 58% of less tumor volume, respectively, for ROS1 siRNA-loaded SSN and Na-Glc-SSN. Na-Glc-SSN had too small particles (according to the size distribution) which might cause rapid elimination of the particles from the body through kidneys (renal clearance), as seen in the biodistribution study. The biodistribution studies of the particles after 4 hours of treatment showed Na-Glc-SSNs had less off-target distribution than SSNs and after 24 hours all siRNAs were cleared from all major organs except tumors. In addition, the mouse plasma binding analysis by Q-TOF LC-MS/MS (new data) revealed less protein binding to smaller particles of Na-Glc-SSNs. Therefore, although SSNs-ROS1 was more prominent in tumor regression than Na-Glc-SSN-ROS1, the tumor-specificity and thus the overall safety profile is much better for the Na-Glc-SSN than the SSN.
2. The description in the abstract is not clear enough, please make deep discussion.
Response: we have discussed accordingly.
3. In the section “Cellular uptake of fluorescence-labeled siRNA”, MCF-7 cells were only treated with siRNA nanoparticle for 4 hours. Please extend the time to compare the uptake of siRNA.
Response: We have added the cellular uptake study of fluorescence-labelled siRNA after 12 hours of treatment (Fig. 11-B and Fig. 12-B).
4. In the Fig. 11(A), “cells treated with free siRNA did not show any fluorescence signals”, please give a reasonable explanation.
Response: Free siRNAs inefficiently enter into cell due to the repulsion between their negatively charged phosphate backbone and the cell membrane which is also negatively charged due to the presence of sulfate-carrying proteoglycans, resulting in no fluorescence signal. Nanoparticles coupled siRNA could bypass the charge barrier of cellular membrane and enter into the cell through endocytosis.
5. Please add images of the biodistribution in vivo.
Response: We have added the images of tumor regression (Fig. 17-B).
6. Most of the image pixels in this manuscript are too low to be identified. Please make further modifications.
Response: We have improved the images pixels.
7. The grammar of the manuscript needs to be modified.
Response: We have checked and corrected.
8. Please make sure the punctuation and abbreviation are accurate and proper.
Response: We have corrected.
9. Some references are outdated. The reference lists should be updated with the recently published studies in the latest three years.
Response: We have corrected.
Reviewer 2 Report
The manuscript by Karim et al. describes the development of strontium sulfite as an agent for siRNA delivery. The manuscript contains a lot of data and is publishable upon addressing the following comments.
1) Major concern is the lack of non-targeting scrambled siRNA control throughout (in vitro and in mice) and lack of material characterization (size/charge/etc.) of the final material containing siRNA. siRNA interacts with the particle and may form bigger size, and thus it's important to understand the property of the final material.
2) The in vitro efficacy (viability) of siRNA is quite minimal and there is no confirmation of siRNA knockdown activity (as mRNA or protein).
3) The material is very pH sensitive. What if siRNA gets released in the tumor environment (that is also acidic), but outside the cells? siRNA will not get to cytosol where it could function. Please comment and include in the manuscript.
4) Is the concentration of NaCl and glucose to control the size of SrSO3 particles physiologically relevant? Is the stabilization effect reversible, meaning whether or not NaCl-stabilized particles will aggregate when introduced to low-salt condition?
5) For Fig. 4, why do authors choose to present particle distribution by number or volume, as opposed to more commonly used intensity mode?
6) The statement that "SSNs prepared in the absence of NaCl and glucose with bigger size seemed to adhere to cell membrane rather than going into the cells" is not convincing from the data in Fig. 11B. Better resolution image may be more convincing.
Author Response
Reviewer #2:
Comments and Suggestions for Authors
The manuscript by Karim et al. describes the development of strontium sulfite as an agent for siRNA delivery. The manuscript contains a lot of data and is publishable upon addressing the following comments.
1. Major concern is the lack of non-targeting scrambled siRNA control throughout (in vitro and in mice) and lack of material characterization (size/charge/etc.) of the final material containing siRNA. siRNA interacts with the particle and may form bigger size, and thus it's important to understand the property of the final material.
Response: Based on our available data (not shown in the manuscript considering the volume of the data already presented in the current manuscript) siRNA complexation with particles slightly increase the overall complex size. We used scrambled siRNA as a control in our previously published papers, showing no antitumor effects for both in vitro and in vivo.(1) We have characterized the particles in terms of particle size by using Zeta Sizer, size and surface morphology by SEM, elemental analysis, surface charge or zeta potential, FT-IR, drug binding affinity and also protein binding by Q-TOF LC-MS/MS.
2.The in vitro efficacy (viability) of siRNA is quite minimal and there is no confirmation of siRNA knockdown activity (as mRNA or protein).
Response: In accordance with our previously published data, we got minimal effect in vitro efficacy but observed remarkable tumor regression effect in mice with siRNA-loaded nanoparticles. We have used siRNAs which were functionally validated by qRT-PCT with more than 80% knockdown efficacy. The LCMS and biodistribution study showed less protein binding and significant tumor accumulation of siRNA into target tumor site, rationalizing the strong antitumor effects of SSNs-siRNA against breast tumor compared to the free SSNs. On the other hand, it is quite established that naked siRNAs of therapeutic importance do not have any anti-cancer effect, since they are degraded following intravenous injection and unable to reach the cytoplasm of the target cells.
3) The material is very pH sensitive. What if siRNA gets released in the tumor environment (that is also acidic), but outside the cells? siRNA will not get to cytosol where it could function. Please comment and include in the manuscript.
Response: We have included the following paragraph in the text, addressing the concern of particle dissolution in tumor extracellular environment:
Since tumor extracellular environment is acidic, dissolution of the pH-sensitive particles could not be ruled out, with the consequence of siRNA release from the particles before their internalization into the target cells. We assume that under that acidic environment, particles could be dissolved either partially or completely depending on the number of particles accumulated there. If the particles are partially dissolved, siRNAs released thereby might enter the target tumor cells by passive diffusion which is rather inefficient considering the size of siRNA and the electrostatic repulsion between the negatively charged phosphate backbone of siRNA and the cell membrane which is also negatively charged due to the presence of sulfate-carrying proteoglycans, while the intact particles with electrostatically associated siRNA would enter via endocytosis resulting in particle dissolution under endosomal acidic pH and siRNA release from endosomes to cytosol. On the other hand, if the particles are fully dissolved in extracellular compartment, the released siRNA would enter the tumor cells exclusively via passive diffusion with the consequence of poor therapeutic outcome. Since we have observed robust anti-tumor effect, we presume that the particles were partially dissolved or remained resistant to degradation under the mild acidic environment.
4) Is the concentration of NaCl and glucose to control the size of SrSO3 particles physiologically relevant? Is the stabilization effect reversible, meaning whether or not NaCl-stabilized particles will aggregate when introduced to low-salt condition?
Response: Yes, size reduction capacity of NaCl and glucose is physiologically relevant. We have used small amount of NaCl and glucose which would be further diluted in systemic circulation. We have optimized the minimum required amount of salts to get maximum amount of SSNs within desirable nanosize range. Since particles are immediately coated by serum proteins as shown by newly included study analyzing spontaneously formed protein corona on nanoparticle surface by Q-TOF LC-MS/MS (Table 1 and 2, Fig. 15), it is unlikely that particles would aggregate after being exposed to blood with the result of dilution of NaCl and glucose.
5) For Fig. 4, why do authors choose to present particle distribution by number or volume, as opposed to more commonly used intensity mode?
Response: We have now added particle size distribution by intensity mode.
6) The statement that "SSNs prepared in the absence of NaCl and glucose with bigger size seemed to adhere to cell membrane rather than going into the cells" is not convincing from the data in Fig. 11B. Better resolution image may be more convincing.
Response: We have faced difficulty to clearly differentiate using optical microscopy the fraction of fluorescence labeled siRNAs-loaded particles being associated with the membrane and the fraction being internalized. However, it is clear from the enlarged image that cell membrane is much brighter for SSNs than Na-Glc-SSNs.
Reviewer 3 Report
Karim et al. propose that inorganic Nanocarrier Strontium Sulfite is effective and efficient in delivering therapeutic siRNAs to cancer cells. This article is well written and scientifically sound. They also demonstrated a new method to synthesis SSNs and siRNA-SSNs complexes. In the methods section they have explained everything in detail. However, they lack experimental controls and molecular mechanistic data. The author has to consider following the comments below,
· According to the author in-organic nanocarriers are better than their organic counterparts. Can the author perform an experiment to show efficiency in-organic NCs compared to Organic NCs.
· Strontium Sulfite NPs are not target specific, As nanoparticles injected twice within three days via IV. Can the author be certain that the desired effects are seen only in breast cells not in another cell type. The author should address off-target or non-specific target by Nanoparticles.
· Successful uptake of SSN-siRNA was confirmed by reduction of tumor size, there is no proof provided by the author that only because of siRNA treatment there is a reduction in the tumor size. The author has provided molecular level data either at mRNA level or protein level to confirm that reduction in tumor size is due to SSN-siRNA treatment in balb-c mice.
· The author has to provide data either at mRNA level or protein level in other organs (kidney, heart, adrenal glands, liver) to prove the off-target site of SSN-siRNA particle.
Author Response
Reviewer #3:
Comments and Suggestions for Authors
Karim et al. propose that inorganic Nanocarrier Strontium Sulfite is effective and efficient in delivering therapeutic siRNAs to cancer cells. This article is well written and scientifically sound. They also demonstrated a new method to synthesis SSNs and siRNA-SSNs complexes. In the methods section they have explained everything in detail. However, they lack experimental controls and molecular mechanistic data. The author has to consider following the comments below,
· According to the author in-organic nanocarriers are better than their organic counterparts. Can the author perform an experiment to show efficiency in-organic NCs compared to Organic NCs?
Response: Iorganic nanocarriers are better than their organic counterparts particularly considering their unique physicochemical properties in addition to those commonly shared by both organic and inorganic particles. For example, as shown in this paper, SSNs are rapidly biodegradable with a small change in pH, a unique feature contributing to fast intracellular release of therapeutic molecules
· Strontium Sulfite NPs are not target specific, as nanoparticles injected twice within three days via IV. Can the author be certain that the desired effects are seen only in breast cells not in another cell type. The author should address off-target or non-specific target by Nanoparticles
.
Response: We have carried out biodistribution analysis which shows high selective accumulation of SSNs and NaCl-Glc-SSNs 24 hrs after intravenous injection of fluorescence siRNA-loaded nanoparticles. Even though we have not coated the particle surface with a tumor-specific ligand, depending on the size nanoparticles could be selectively accumulated into the tumors due to enhanced permeability and retention effect (EPR) of tumors which possess highly leaky vasculature and underdeveloped lymphatic drainage system. Moreover, particles have been shown to be immediately coated by serum proteins (newly included study analyzing spontaneously formed protein corona on nanoparticle surface by Q-TOF LC-MS/MS as shown in Table 1 and 2, Fig. 15)), apparently contributing tumor selectivity of the nanoparticles.
· Successful uptake of SSN-siRNA was confirmed by reduction of tumor size, there is no proof provided by the author that only because of siRNA treatment there is a reduction in the tumor size. The author has provided molecular level data either at mRNA level or protein level to confirm that reduction in tumor size is due to SSN-siRNA treatment in balb-c mice.
The author has to provide data either at mRNA level or protein level in other organs (kidney, heart, adrenal glands, liver) to prove the off-target site of SSN-siRNA particle.
Response: We have used siRNAs which were functionally validated by qRT-PCT with more than 80% knockdown efficacy. The LCMS and biodistribution study showed less protein binding and significant tumor accumulation of siRNA into target tumor site, rationalizing the strong antitumor effects of SSNs-siRNA against breast tumor compared to the free SSNs. On the other hand, it is quite established that naked siRNAs of therapeutic importance do not have any anti-cancer effect, since they are degraded following intravenous injection and unable to reach the cytoplasm of the target cells.
Reviewer 4 Report
The manuscript "Strontium sulfite: a new pH-responsive inorganic nano-carrier to deliver therapeutic siRNAs to cancer cells" is well written and easy to understand. The Authors describe how the use of strontium sulfite, NaCl and Glucose can facilitate and increase the entry of siRNA into the cancer cells.
For me the work is suitable for the publication in Pharmaceutics, after little revisions.
In Materials and Methods the authors should standardize the font size.
In Figura 2A the legend is superimposed on the histogram.
Lane 392: the authors should delete the comma and add the appropriate unit for adsorbance.
Fig. 11 A. I don't know if the microscope resolution is not good but also the cells treated with media (untreated cells) have a shape quite different from the classical MCF-7 cells. How is this possible? Can the authors improve the images?
In Fig 10A, the images of cells treated with Glucose-SSN BF seems not exactly corresponding to the respective IF image. May the Authors perform merged images for all the treatments?
Author Response
Reviewer #4:
Comments and Suggestions for Authors
The manuscript "Strontium sulfite: a new pH-responsive inorganic nano-carrier to deliver therapeutic siRNAs to cancer cells" is well written and easy to understand. The Authors describe how the use of strontium sulfite, NaCl and Glucose can facilitate and increase the entry of siRNA into the cancer cells.
For me the work is suitable for the publication in Pharmaceutics, after little revision
In Materials and Methods, the authors should standardize the font size.
Response: we have corrected the font size.
In Figure 2A the legend is superimposed on the histogram.
Response: We have corrected.
Line 392: the authors should delete the comma and add the appropriate unit for adsorbance.
Response: Corrected
Fig. 11 A. I don't know if the microscope resolution is not good but also the cells treated with media (untreated cells) have a shape quite different from the classical MCF-7 cells. How is this possible? Can the authors improve the images?
Response: We have improved the images.
In Fig 10A, the images of cells treated with Glucose-SSN BF seems not exactly corresponding to the respective IF image. May the Authors perform merged images for all the treatments?
Response: We have corrected.
Reviewer 5 Report
Authors did a great job formulating Strontium Sulfite nano-system with NaCl and Glucose as additional excipients which enhances the stabilization of this formulation for the successful siRNA delivery to breast cancer cells and tumors.
These nanoparticles show excellent physico-chemical properties and in vitro and in vivo profiles which is suggestive of superior delivery of desired siRNA.
Author Response
Reviewer #5:
Authors did a great job formulating Strontium Sulfite nano-system with NaCl and Glucose as additional excipients which enhances the stabilization of this formulation for the successful siRNA delivery to breast cancer cells and tumors.
These nanoparticles show excellent physico-chemical properties and in vitro and in vivo profiles which is suggestive of superior delivery of desired siRNA
Response: Thank you very much for your comments.
Reviewer 6 Report
In this manuscript Karim et. al. describe the preclinical development of strontium sulfite nanoparticles (SSN) for cancer gene therapy applications. Charge-charge interactions are used to load SSN with negatively charged siRNA for gene therapy applications. Particles are tested for activity in vitro against cancer cells and in vivo using an animal model of cancer to demonstrate tumor volume reduction. The manuscript is confusing to read and understand with some key controls missing in some experiments.
Some comments to improve the manuscript are included below
1. It is unclear based on reading this manuscript how the body degrades and eliminates this inorganic material from the body to avoid chronic toxicity. Lower pH found in endosomes can break the particles down but its unlcear how the degradation products are eliminated. Strontium ions could easily replace ions like Ca, Mg and Na from the body and could be toxic. Further per, the authors findings, these particles could easily aggregate in the body and this could lead to toxic effects too. Please comment with support from literature references.
2. If the mechanism proposed in figure 3 is indeed the actual mechanism of particle aggregation and dispersion, why doesn't the zeta potentials in Figure 2B support this hypothesis? Zeta Potential stays almost the same with and without addition of glucose and/or NaCl. Zeta Potentials for all particles included in Figure 2A should be provided in Figure 2B.
3. Representative images of particles that support the proposed mechanism in Figure 3 are required. Combining Figure 3 and 5 in a coherent way may be useful. However, SEM requires the particles be dried under vacuum for imaging and as such cannot provide conclusive evidence that aggregation is not an artifact of the sample preparation methods. This reviewer can;t see any particles under 100nm in any of the images in Fig 5.
4. The value of Figure 4 is unclear. Please consider moving it to supplementary information.
5. Absorbances going down under acidic pH is not proof enough that SSNs are being dissolved. Please provide additional information.
6. The efficacy of the siRNA loaded particles in vitro is not impressive.No SSN reduces viability of cells below 60%. Please comment. Were scrambled siRNA sequences tested in the free and loaded form?
7.Why were biodistribution studies not conducted for all organs at 4h and 24h after injection? Figure 14 includes the 4 and 24h time points only for the tumor.
7. No treatment control is missing in Figure 15. Please include. Why was the EGFR siRNA not evaluated in vivo?
8. Conclusions need to be rewritten to discuss future steps in the development of SSN particles, especially their ADME profiles.
Minor
- Scale bar in Figure 1B is not legible. Are these results from the electron microscopy studies? A higher resolution and zoomed in version of Figure 1B is needed to verify claims in Figure 3.
- Is the Olympus Microscope CKX41 an electron microscope? Additional details are needed int he methods.
- Font type and sizes change suddenly in the manuscript which is distracting.
- Units of particle size diameter in Figure 2A is missing.
- Figure 15 - The images are not in the order the figure legend states they are.
- In general, the authors need to carefully look at their presentation of results to improve readability.
Author Response
Reviewer #6:
Comments and Suggestions for Authors
In this manuscript Karim et. al. describes the preclinical development of strontium sulfite nanoparticles (SSN) for cancer gene therapy applications. Charge-charge interactions are used to load SSN with negatively charged siRNA for gene therapy applications. Particles are tested for activity in vitro against cancer cells and in vivo using an animal model of cancer to demonstrate tumor volume reduction. The manuscript is confusing to read and understand with some key controls missing in some experiments.
Some comments to improve the manuscript are included below
1.It is unclear based on reading this manuscript how the body degrades and eliminates this inorganic material from the body to avoid chronic toxicity. Lower pH found in endosomes can break the particles down but its unlcear how the degradation products are eliminated. Strontium ions could easily replace ions like Ca, Mg and Na from the body and could be toxic. Further per, the authors findings, these particles could easily aggregate in the body and this could lead to toxic effects too. Please comment with support from literature references.
Response: We have used very small amount of Sr(2+) salt to prepare SSNs which would be further diluted in large amount of blood. We hypothesized that the diluted SSNs could easily be degraded in endosomes, excluded from the cells via cell membrane-associated cation pumps and eliminated from the body through renal clearance, without any cytotoxicity as shown in Fig. 17(C). Furthermore, upon intravenous administration, the individual particles would interact with serum proteins that could prevent particle-particle aggregation. We have included a new sstudy analyzing spontaneously formed protein corona on nanoparticle surface by Q-TOF LC-MS/MS as shown in Table 1 and 2, Fig. 15.
2.If the mechanism proposed in figure 3 is indeed the actual mechanism of particle aggregation and dispersion, why doesn't the zeta potentials in Figure 2B support this hypothesis? Zeta Potential stays almost the same with and without addition of glucose and/or NaCl. Zeta Potentials for all particles included in Figure 2A should be provided in Figure 2B.
Response: The presence of glucose caused a reduction of particle aggregation by lying between the particles without directly interacting with the particles, while Na and Cl ions temporally binds with the particle surface with approximately same ratio of positive (Na) and negative (Cl) ions. Therefore, the zeta potential for both modified and unmodified SSNs were almost same.
3. Representative images of particles that support the proposed mechanism in Figure 3 are required. Combining Figure 3 and 5 in a coherent way may be useful. However, SEM requires the particles be dried under vacuum for imaging and as such cannot provide conclusive evidence that aggregation is not an artifact of the sample preparation methods. This reviewer can;t see any particles under 100nm in any of the images in Fig 5.
Response: Fig. 5 clearly shows the size of an individual particle of SSN or NaCl-Glc-SSN. Sample preparation method used SEM causes aggregation of individual particles, which is also visible in the same figure. Since the particles below 100 nm constitute the minor factions of the particles, we hardly see them under SEM.
4. The value of figure 4 is unclear. Please consider moving it to supplementary information.
Response: We have corrected accordingly.
5. Absorbances going down under acidic pH is not proof enough that SSNs are being dissolved. Please provide additional information.
Response: We have published earlier showing that gradual decrease in turbidity with decreasing pH is perfectly correlated with the release of drugs from the particles (2).
6. The efficacy of the siRNA loaded particles in vitro is not impressive. No SSN reduces viability of cells below 60%. Please comment. Were scrambled siRNA sequences tested in the free and loaded form?
Response: According to our published papers, we got quite minimal in vitro efficacy with potentially therapeutic siRNAs unlike small molecule anti-cancer drugs, but the former demonstrated remarkable tumor regression effect in mice model study (1,3,4)
7. Why were biodistribution studies not conducted for all organs at 4h and 24h after injection? Figure 14 includes the 4 and 24h time points only for the tumor.
Response: We have added the biodistribution for all organs at 4 and 24 hrs.
7. No treatment control is missing in Figure 15. Please include. Why was the EGFR siRNA not evaluated in vivo?
Response: In our lab we have consistently found that no treatment group and only NPs groups show the same pattern of tumor growth and that’s why we have used only NPs as our control group (1, 3, 4). We have tested the ROS1 and EGFR before both in vitro and in vivo in our lab (1, 3). Our main aim here is to evaluate the ability of SSNs to carry siRNA into target tumor site. Since we previously obtained good antitumor effect by using ROS1 siRNA-loaded carbonate apatite, we have used here the same siRNA for intravenous delivery using SSNs and NaCl-Glc-SSNs.
8. Conclusions need to be rewritten to discuss future steps in the development of SSN particles, especially their ADME profiles.
Response: We have corrected
Minor
- Scale bar in Figure 1B is not legible. Are these results from the electron microscopy studies? A higher resolution and zoomed in version of Figure 1B is needed to verify claims in Figure 3.
Response: we have corrected.
- Is the Olympus Microscope CKX41 an electron microscope? Additional details are needed into the methods.
Response: No, it is not electron microscope.
- Font type and sizes change suddenly in the manuscript which is distracting.
Response: we have corrected.
- Units of particle size diameter in Figure 2A is missing.
Response: Units have been added.
- Figure 15 - The images are not in the order the figure legend states they are.
Response: Rearranged according to the legend states.
- In general, the authors need to carefully look at their presentation of results to improve readability.
Response: We have tried to improve it.
1. Tiash S, Kamaruzman NIB, Chowdhury EH. Carbonate apatite nanoparticles carry siRNA (s) targeting growth factor receptor genes egfr1 and erbb2 to regress mouse breast tumor. Drug delivery. 2017;24(1):1721-30.
2. Kamaruzman NI, Tiash S, Ashaie M, Chowdhury EH. siRNAs Targeting Growth Factor Receptor and Anti-Apoptotic Genes Synergistically Kill Breast Cancer Cells through Inhibition of MAPK and PI-3 Kinase Pathways. Biomedicines. 2018 Jun 22;6(3).
3. Mehbuba Hossain S, Chowdhury EH. Citrate- and Succinate-Modified Carbonate Apatite Nanoparticles with Loaded Doxorubicin Exhibit Potent Anticancer Activity against Breast Cancer Cells. Pharmaceutics. 2018 Mar 11;10(1).
4. Tiash S, Chowdhury EH. siRNAs targeting multidrug transporter genes sensitize breast tumor to doxorubicin in a syngeneic mouse model. J Drug Target. 2018 Sep 17:1-49.
Round 2
Reviewer 1 Report
The revision is ready for publication.
Author Response
Reviewer #1:
The revision is ready for publication
Response: Thank you very much.
Reviewer 2 Report
The fact that SSN works better in vivo than Na-Glc-SSN should be included in the abstract as well. Also, please also mention why ROS1 siRNA was selected over EGFR siRNA in the abstract.
Lastly, data on scrambled siRNA on the same nanoparticles should be included as supplementary data. Otherwise, readers may question the validity of the results presented herein.
Author Response
Reviewer #2:
The fact that SSN works better in vivo than Na-Glc-SSN should be included in the abstract as well. Also, please also mention why ROS1 siRNA was selected over EGFR siRNA in the abstract.
Lastly, data on scrambled siRNA on the same nanoparticles should be included as supplementary data. Otherwise, readers may question the validity of the results presented herein.
Response:
We have edited the abstract accordingly (labeled in purple colour):
“ROS1 siRNA with potential therapeutic role in treating 4T1-induced breast tumor was selected for subsequent in vivo tumor regression study, revealing that ROS1 siRNA-loaded SSNs exerted more significant anti-tumor effects than Na-Glc-SSNs carrying the same siRNA following intravenous administration, without any systemic toxicity.”
We used scrambled siRNA as a control in our previously published papers, showing no antitumor effects for both in vitro and in vivo.(1) We have used siRNAs which were functionally validated by qRT-PCT with more than 80% knockdown efficacy.
1.Tiash S, Kamaruzman NIB, Chowdhury EH. Carbonate apatite nanoparticles carry siRNA (s) targeting growth factor receptor genes egfr1 and erbb2 to regress mouse breast tumor. Drug delivery. 2017;24(1):1721-30.
Reviewer 3 Report
The Author has answered all my question. This paper is ready for publication
Author Response
Reviewer #3:
The Author has answered all my question. This paper is ready for publication
Response: Thank you very much.
Reviewer 6 Report
The authors have tried to address the comments and the work is ready for publication with some minor fixes to improve readability.
Author Response
Reviewer #6:
The authors have tried to address the comments and the work is ready for publication with some minor fixes to improve readability.
Response: Thank you very much. We have fixed the minor mistakes.